# Experience-dependent evolution of odor mixture representations in piriform cortex

**Alice Berners-Lee**[1☯], **Elizabeth Shtrahman**[1☯], **Julien Grimaud**[1,2], **Venkatesh N. Murthy**[1]*

**1** Department of Molecular and Cellular Biology and Center for Brain Science, Harvard University, Cambridge, Massachusetts, United States of America, **2** Cell Engineering Laboratory (CellTechs), Sup'Biotech, Villejuif, France

☯ These authors contributed equally to this work.
* vnmurthy@fas.harvard.edu

**Data Availability Statement:** The data are available for download from https://datadryad.org/stash/share/bC3NdXWDIlJZYrRtq60q0WDQYhjZZuLulv91dm9WcYU.

## Abstract

Rodents can learn from exposure to rewarding odors to make better and quicker decisions. The piriform cortex is thought to be important for learning complex odor associations; however, it is not understood exactly how it learns to remember discriminations between many, sometimes overlapping, odor mixtures. We investigated how odor mixtures are represented in the posterior piriform cortex (pPC) of mice while they learn to discriminate a unique target odor mixture against hundreds of nontarget mixtures. We find that a significant proportion of pPC neurons discriminate between the target and all other nontarget odor mixtures. Neurons that prefer the target odor mixture tend to respond with brief increases in firing rate at odor onset compared to other neurons, which exhibit sustained and/or decreased firing. We allowed mice to continue training after they had reached high levels of performance and find that pPC neurons become more selective for target odor mixtures as well as for randomly chosen repeated nontarget odor mixtures that mice did not have to discriminate from other nontargets. These single unit changes during overtraining are accompanied by better categorization decoding at the population level, even though behavioral metrics of mice such as reward rate and latency to respond do not change. However, when difficult ambiguous trial types are introduced, the robustness of the target selectivity is correlated with better performance on the difficult trials. Taken together, these data reveal pPC as a dynamic and robust system that can optimize for both current and possible future task demands at once.

## Introduction

Rodents depend on odor learning for many important behaviors [1,2]. The piriform cortex, a key olfactory cortical region, clearly reports single odors with passive exposure, with about 10% to 20% of piriform neurons encoding a single odor [3–6]. The piriform cortex, made up of the anterior piriform cortex (aPC) and posterior piriform cortex (pPC), is thought to be important in associative learning [7–11]. Indeed, when a single odor is paired with reward, although pPC neurons have not been found to encode odor valence [12,13], more aPC neurons are often recruited to respond to those odors compared to unrewarded odors [14,15]. In the pPC specifically, more neurons are recruited to distinguish between 2 single odors that can lead to obtaining reward or avoiding punishment [16].

**Funding:** This work was supported in part by the National Institutes of Health (grant number R01DC016289). The funders had no role in study design, data collection and analysis, decision to publish, or preparation of the manuscript.

**Competing interests:** The authors have declared that no competing interests exist.

However, in a rodent's natural environment, odors can be more complex and there may be hundreds of overlapping odors that need to be distinguished from a goal or target odor. Mice can detect the presence of a single odor within a large odor mixture [17], in a go/no-go design as well as in a left-right alternate choice design [18]. To investigate how pPC representations change with experience, we developed a complex odor task where mice learned to categorize 1 target odor mixture as different from all the hundreds of other odor mixtures. After a quick learning phase, we recorded from pPC neurons. We describe how pPC neurons encode odor cues in the task and then observe how that representation changes with overtraining. We observed that the pPC overrepresents important odor mixtures and continues to do so even without reaping any immediate benefit in reward rate. Finally, we introduced "probe" trials that challenged the mouse with ambiguous odor mixtures and observed how the representation in pPC leverages its robust coding to overcome the novel challenge trials. Taken together, our data reveal a robust and constantly dynamic pPC, which can flexibly prepare for the possible future.

## Results

### Mice learn to identify a unique target odor mixture

Four head-fixed water-restricted male mice were presented with an odor mixture made up of 3 odors (Fig 1). Mice were trained to lick left for the unique target odor mixture (Ethyl tiglate, Allyl tiglate, and Methyl tiglate) and right for any other combination of odors (Fig 1A and 1B). The nontarget odor mixtures also contained 3 single odor components randomly drawn from 13 odors. In each trial, the odor was presented for 2 seconds and a lick to either port in the 2,500 ms after odor onset was counted as a response and rewarded for correct responses (Fig 1D). Mice learned the task within a few days ("Learning"; Fig 1E; Table 1, row 2). In separate cohorts of mice, other target odor mixtures were used, and mice were similarly adept at performing the task, arguing for the generality of our results (S1 Fig). After this initial learning, mice continued to perform the task ("Overtraining"; Fig 1F; days 8 to 28; Table 1, row 3). In order to compare neural responses to learned (target) and irrelevant mixtures, during these sessions, some of the nontarget trials were repeated multiple times ("nontarget repeats"; Fig 1B). In later sessions, probe mixtures were introduced (Fig 1C and 1G; days 22 to 38; Table 1, row 4), where a mixture contained one of the 3 odors from the target mixture along with 2 randomly chosen nontarget odors. Mice were rewarded for classifying both probe and nontarget repeat trials as nontarget mixtures (using the criterion that all 3 components must be present for it to be a target odor). Mice improved their accuracy in nontarget and target trials across sessions only in the learning stage of the task, not in later stages where they appear to be consistently at ceiling (significant interaction between stage and session, Table 1, row 1; session is significant in the learning and probe stages but not the overtraining stage, Table 1, rows 2 to 4). These data show that mice can quickly learn to tell apart a target odor mixture from hundreds of other nontarget mixtures and further learn to place ambiguous odor mixtures in the correct category.

### Piriform neurons show brief upward or prolonged downward modulations at odor onset

Tetrodes were implanted in the pPC (0.5 mm posterior and 3.8 mm lateral from bregma, and 3.8 mm ventral from the brain surface; S2 Fig) and spikes were clustered to isolate single putative pyramidal neurons (S3 Fig). We found that the majority of neurons are modulated at odor onset (Fig 2). We also found that some neurons respond to the first lick the mouse made

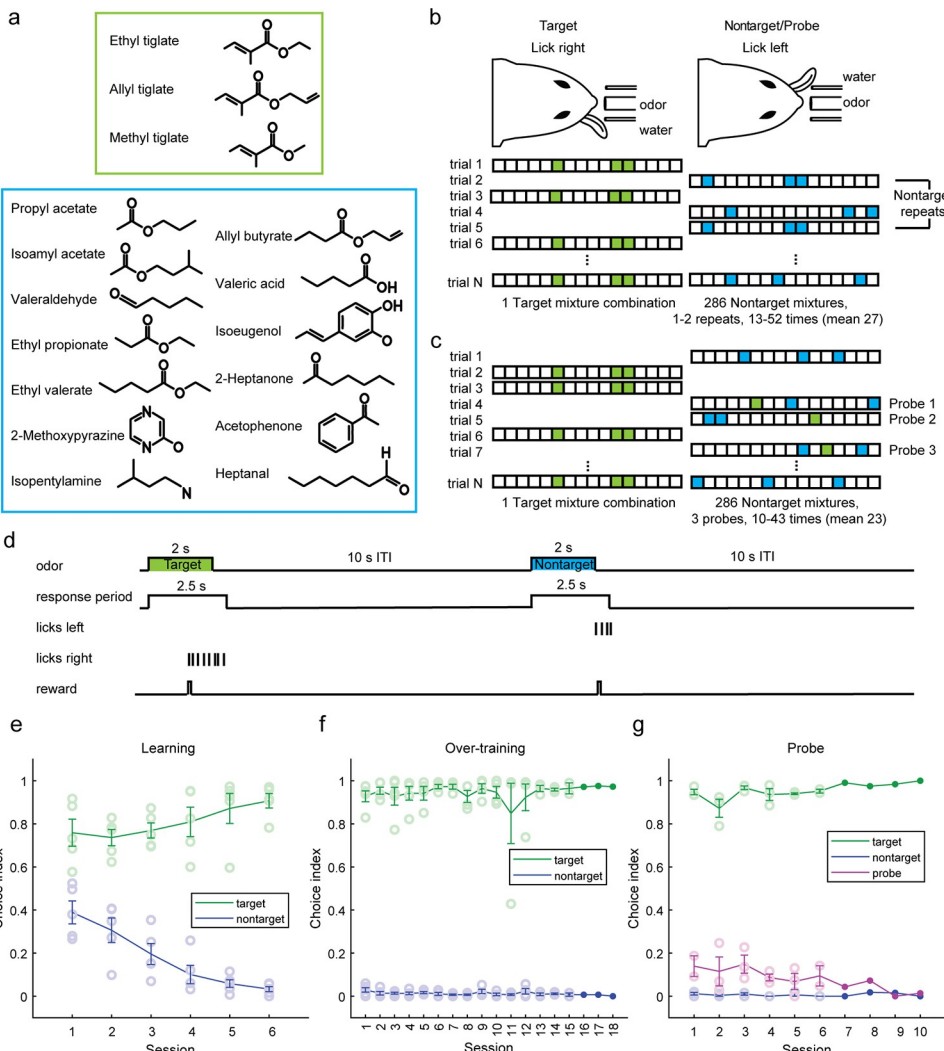

**Fig 1. Task and behavior.** (**a**) Odor panel. Three odor components make up the target mixture (top), and the nontarget odors are drawn from 13 other components (bottom). Each nontarget trial consists of 3 of these 13 components. (**b**) Behavioral setup (top) and stimulus structure (bottom). Colored boxes mark the odorants present in each trial (green for target, blue for nontarget). Some nontarget trials were repeated, while others only appeared once. Trial types are pseudorandomly interleaved. (**c**) Probe component mixture task. Some of the nontarget trials are probe component mixtures containing 1 target odorant and 2 nontarget odorants. Mice receive water rewards for categorizing probe mixtures as nontargets (left lick choice). (**d**) Trial structure. Example trial structure for a correct target and nontarget trial. There was a 2-second odor duration followed by a 10-second intertrial interval (ITI). The response period was the entirety of the odor duration and the following 500 ms. (**e**-**g**) Target/nontarget task performance of 4 mice for multiple sessions. Choice index is 1 for lick right, and 0 for lick left. Sessions arranged in order mice performed them. Mice change their choices for target and nontarget trials across sessions during the Learning stage but not following 2 stages (see Table 1, rows 1–4). Error bars represent mean +/− SEM across mice, open circles represent each session. (**e**) Learning phase, with only target and nontarget trials (no electrophysiology recordings during this time). Session 1 for each mouse is the first session with blocked trials (see Materials and methods for more details on training). (**f**) Overtraining. Session 1 is the first session with random trial structure. (**g**) Sessions with probe trials. Session 1 is when probes were first introduced. The underlying data for this figure are available for download from https://datadryad.org/stash/share/bC3NdXWDllJZYrRtq60q0WDQYhjZZuLulv91dm9WcYU.

(S4 Fig; neurons with such responses were excluded from counting as odor onset modulated neurons) as well as a small proportion that respond to odor offset (S5 Fig). We focused subsequent analyses on the largest portion of task-relevant neurons, those modulated at odor onset.

**Table 1. Statistics of comparisons.**

| Row 1: Fig 1E–1G, ANOVA with interaction | Source | Sum Sq. | d.f. | Mean Sq. | F | Prob>F |
|---|---|---|---|---|---|---|
| | Session | 0.358134 | 1 | 0.358134 | 69.027 | $1.05 \times 10^{-14}$ |
| | Mice | 0.059799 | 4 | 0.01495 | 2.881411 | 0.023577 |
| | T/NT | 0.075175 | 1 | 0.075175 | 14.4892 | 0.000183 |
| | Stage | 0.993275 | 2 | 0.496638 | 95.72229 | $1.58 \times 10^{-30}$ |
| | Session*Stage | 0.485656 | 2 | 0.242828 | 46.8028 | $1.26 \times 10^{-17}$ |
| | Error | 1.125865 | 217 | 0.005188 | | |
| | Total | 2.799787 | 227 | | | |
| Row 2: Fig 1E, Learning, ANOVA | Source | Sum Sq. | d.f. | Mean Sq. | F | Prob>F |
| | Session | 0.516392 | 1 | 0.516392 | 50.09605 | $3.38 \times 10^{-9}$ |
| | Mice | 0.124924 | 4 | 0.031231 | 3.029781 | 0.025306 |
| | T/NT | 0.001743 | 1 | 0.001743 | 0.169131 | 0.682543 |
| | Error | 0.546326 | 53 | 0.010308 | | |
| | Total | 1.189385 | 59 | | | |
| Row 3: Fig 1F, Overlearning, ANOVA | Source | Sum Sq. | d.f. | Mean Sq. | F | Prob>F |
| | Session | 0.000181 | 1 | 0.000181 | 0.050251 | 0.823012 |
| | Mice | 0.035125 | 4 | 0.008781 | 2.444099 | 0.050292 |
| | T/NT | 0.061885 | 1 | 0.061885 | 17.22453 | $6.27 \times 10^{-5}$ |
| | Error | 0.427546 | 119 | 0.003593 | | |
| | Total | 0.525476 | 125 | | | |
| Row 4: Fig 1G, Probe, ANOVA | Source | Sum Sq. | d.f. | Mean Sq. | F | Prob>F |
| | Session | 0.004692 | 1 | 0.004692 | 4.609343 | 0.03842 |
| | Mice | 0.000652 | 2 | 0.000326 | 0.320178 | 0.728011 |
| | T/NT | 0.024976 | 1 | 0.024976 | 24.53755 | $1.63 \times 10^{-5}$ |
| | Error | 0.037661 | 37 | 0.001018 | | |
| | Total | 0.068314 | 41 | | | |
| Row 5: Fig 2E, ANOVA | Source | Sum Sq. | d.f. | Mean Sq. | F | Prob>F |
| | UpDown | 1.054443 | 1 | 1.054443 | 33.01469 | $1.12 \times 10^{-8}$ |
| | Mouse | 0.001382 | 3 | 0.000461 | 0.014426 | 0.997635 |
| | Session | 0.000253 | 1 | 0.000253 | 0.007923 | 0.929086 |
| | TimeBin | 0.644067 | 1 | 0.644067 | 20.1658 | $7.69 \times 10^{-6}$ |
| | UpDown*TimeBin | 1.333638 | 1 | 1.333638 | 41.7563 | $1.43 \times 10^{-10}$ |
| | Error | 44.29885 | 1,387 | 0.031939 | | |
| | Total | 46.2904 | 1,394 | | | |
| Row 6: Fig 2F, ANOVA | Source | Sum Sq. | d.f. | Mean Sq. | F | Prob>F |
| | UpDown | 0.353416 | 1 | 0.353416 | 3.953052 | 0.046963 |
| | Mouse | 2.134405 | 3 | 0.711468 | 7.957973 | $2.89 \times 10^{-5}$ |
| | Session | 0.594733 | 1 | 0.594733 | 6.652255 | 0.009995 |
| | TimeBin | 1.750916 | 1 | 1.750916 | 19.58448 | $1.03 \times 10^{-5}$ |
| | UpDown*TimeBin | 4.588381 | 1 | 4.588381 | 51.32233 | $1.21 \times 10^{-12}$ |
| | Error | 137.8598 | 1,542 | 0.089403 | | |
| | Total | 154.974 | 1,549 | | | |
| Row 7: Fig 4A, chi-squared test | chi2stat | d.f. | Observed | | | |
| | 151.4704212 | 1 | 84 | 330 | 202 | 106 |
| | *P* value | | Expected | | | |
| | $8.27124 \times 10^{-35}$ | | 164 | 250.0055 | 122.0055 | 185.9945 |

*(Continued)*

**Table 1.** (Continued)

| Row 8: Fig 4B, ANOVA | Source | Sum Sq. | d.f. | Mean Sq. | F | Prob>F |
|---|---|---|---|---|---|---|
| | UpDown | 8.855995 | 1 | 8.855995 | 148.7132 | $3.69 \times 10^{-24}$ |
| | Mouse | 0.188875 | 3 | 0.062958 | 1.057222 | 0.369238 |
| | Session | 0.181743 | 1 | 0.181743 | 3.051894 | 0.082704 |
| | Error | 8.873076 | 149 | 0.059551 | | |
| | Total | 18.15184 | 154 | | | |
| Row 9: Fig 4C, ANOVA | Source | Sum Sq. | d.f. | Mean Sq. | F | Prob>F |
| | TrialType | 0.810516 | 2 | 0.405258 | 19.71274 | $2.41 \times 10^{-8}$ |
| | Mouse | 0.092191 | 3 | 0.03073 | 1.494796 | 0.218226 |
| | Session | 0.061477 | 1 | 0.061477 | 2.990393 | 0.085777 |
| | Error | 3.145401 | 153 | 0.020558 | | |
| | Total | 4.098207 | 159 | | | |
| Row 10: Fig 4C, Post hoc Bonferroni | Group1 | Group2 | Lower | Estimate difference | Upper | P |
| | Target | Probe | 0.03 | 0.128721 | 0.224955 | 0.004432 |
| | Target | NT repeat | 0.09 | 0.146738 | 0.208162 | $1.2 \times 10^{-7}$ |
| | Probe | NT repeat | −0.1 | 0.018017 | 0.124631 | 1 |
| Row 11: Fig 4F, ANOVA | Source | Sum Sq. | d.f. | Mean Sq. | F | Prob>F |
| | Target/NT | 0.413662 | 1 | 0.413662 | 2.281431 | 0.131243 |
| | Mouse | 2.017658 | 3 | 0.672553 | 3.709268 | 0.01134 |
| | Session | 0.112544 | 1 | 0.112544 | 0.620704 | 0.43097 |
| | TimeBin | 1.863444 | 1 | 1.863444 | 10.27728 | 0.001389 |
| | Target/NT*TimeBin | 1.576046 | 1 | 1.576046 | 8.69222 | 0.003269 |
| | Error | 183.4926 | 1,012 | 0.181317 | | |
| | Total | 189.9753 | 1,019 | | | |
| Row 12: Fig 5A, ANOVA | Source | Sum Sq. | d.f. | Mean Sq. | F | Prob>F |
| | Mice | 0.004093 | 3 | 0.001364 | 2.087038 | 0.103782 |
| | Sessions | 0.000593 | 1 | 0.000593 | 0.907151 | 0.342224 |
| | Trialtype | 0.00013 | 2 | $6.51 \times 10^{-5}$ | 0.099512 | 0.905331 |
| | Error | 0.111133 | 170 | 0.000654 | | |
| | Total | 0.117324 | 176 | | | |
| Row 13: Fig 5B, ANOVA | Source | Sum Sq. | d.f. | Mean Sq. | F | Prob>F |
| | Mouse | 3,561,595 | 3 | 1,187,198 | 32.47128 | $1.21 \times 10^{-16}$ |
| | Session | 8,722.252 | 1 | 8,722.252 | 0.238564 | 0.625875 |
| | TrialType | 433,889.3 | 2 | 216,944.6 | 5.933694 | 0.003228 |
| | Error | 6,215,452 | 170 | 36,561.48 | | |
| | Total | 10,225,968 | 176 | | | |
| Row 14: Fig 5D, ANOVA | Source | Sum Sq. | d.f. | Mean Sq. | F | Prob>F |
| | Session | 0.120362 | 1 | 0.120362 | 6.320851 | 0.014945 |
| | Mouse | 0.045618 | 3 | 0.015206 | 0.79855 | 0.500091 |
| | Error | 1.028268 | 54 | 0.019042 | | |
| | Total | 1.202804 | 58 | | | |
| Row 15: Fig 5E, ANOVA | Source | Sum Sq. | d.f. | Mean Sq. | F | Prob>F |
| | Session | 0.212626 | 1 | 0.212626 | 16.04186 | 0.000191 |
| | Mouse | 0.075496 | 3 | 0.025165 | 1.898622 | 0.140784 |
| | Error | 0.715741 | 54 | 0.013254 | | |
| | Total | 0.962285 | 58 | | | |

(*Continued*)

**Table 1.** (Continued)

| Row 16: Fig 6B, ANOVA | Source | Sum Sq. | d.f. | Mean Sq. | F | Prob>F |
|---|---|---|---|---|---|---|
| | TrialType | 0.838543 | 2 | 0.419272 | 8.505782 | 0.000203 |
| | Trial | $3.65 \times 10^{-6}$ | 1 | $3.65 \times 10^{-6}$ | $7.41 \times 10^{-5}$ | 0.993134 |
| | Session | 9.321488 | 1 | 9.321488 | 189.1055 | $8.91 \times 10^{-43}$ |
| | Mouse | 1.302724 | 3 | 0.434241 | 8.809476 | $7.84 \times 10^{-6}$ |
| | Rewarded | 3.937755 | 1 | 3.937755 | 79.88542 | $4.41 \times 10^{-19}$ |
| | Error | 760.9289 | | 0.049293 | | |
| | Total | 778.6345 | | | | |
| Row 17: Fig 6C top, ANOVA | Source | Sum Sq. | d.f. | Mean Sq. | F | Prob>F |
| | Trials | 0.483832 | 1 | 0.483832 | 10.99275 | 0.000919 |
| | Sessions | 6.976238 | 1 | 6.976238 | 158.5014 | $5.85 \times 10^{-36}$ |
| | Mouse | 1.230924 | 3 | 0.410308 | 9.322271 | $3.79 \times 10^{-6}$ |
| | Error | 310.5169 | 7,055 | 0.044014 | | |
| | Total | 319.948 | 7,060 | | | |
| Row 18: Fig 6C middle, ANOVA | Source | Sum Sq. | d.f. | Mean Sq. | F | Prob>F |
| | Trials | 0.171656 | 1 | 0.171656 | 3.138241 | 0.076538 |
| | Sessions | 2.953633 | 1 | 2.953633 | 53.99885 | $2.33 \times 10^{-13}$ |
| | Mouse | 0.734462 | 3 | 0.244821 | 4.475855 | 0.003828 |
| | Error | 270.3179 | 4,942 | 0.054698 | | |
| | Total | 274.134 | 4,947 | | | |
| Row 19: Fig 6C bottom, ANOVA | Source | Sum Sq. | d.f. | Mean Sq. | F | Prob>F |
| | Trials | 0.00465 | 1 | 0.00465 | 0.084131 | 0.771804 |
| | Sessions | 0.811176 | 1 | 0.811176 | 14.67784 | 0.000131 |
| | Mouse | 0.810941 | 3 | 0.270314 | 4.891193 | 0.002166 |
| | Error | 114.8967 | 2,079 | 0.055265 | | |
| | Total | 116.2065 | 2,084 | | | |
| Row 20: Fig 7B, ANOVA | Source | Sum Sq. | d.f. | Mean Sq. | F | Prob>F |
| | Session | 0.39685 | 1 | 0.39685 | 35.09976 | $3.72 \times 10^{-8}$ |
| | Mice | 0.305844 | 4 | 0.076461 | 6.762656 | $6.7 \times 10^{-5}$ |
| | Probe # | 0.373161 | 2 | 0.18658 | 16.50228 | $5.49 \times 10^{-7}$ |
| | Error | 1.232391 | 109 | 0.011306 | | |
| | Total | 2.08023 | 116 | | | |

Across all sessions, 308 of 1,055 neurons isolated from tetrode recordings (Fig 2A), an average of 29% of neurons a session (Fig 2B), have significantly higher firing in the second after odor onset (Fig 2C and 2D, left two examples) compared to baseline (see Materials and methods). In addition, 414 of 1,055 neurons (Fig 2A), an average of 39% of neurons a session (Fig 2B), have significantly reduced firing in the second after odor onset (Fig 2C and 2D, right example) compared to baseline (see Materials and methods). The proportion of neurons with significantly modulated firing rates is not related to the number of neurons recorded in any given session (S6 Fig, range of 3 to 30 neurons per session, mean = 12.2, median = 13).

We measured the total duration that neurons remain upwardly modulated after odor onset ("Odor-on up") and found the distribution of these durations to be bimodal (Fig 2E, top); many neurons responded with brief modulations, and another set responded with prolonged modulations (Fig 2G, top and middle), most frequently modulated within 200 to 600 ms after odor onset (Fig 2G, top). By contrast, neurons that reduce their firing rates after odor onset ("Odor-on down") tend to have prolonged modulations (Fig 2E, bottom) across most of the

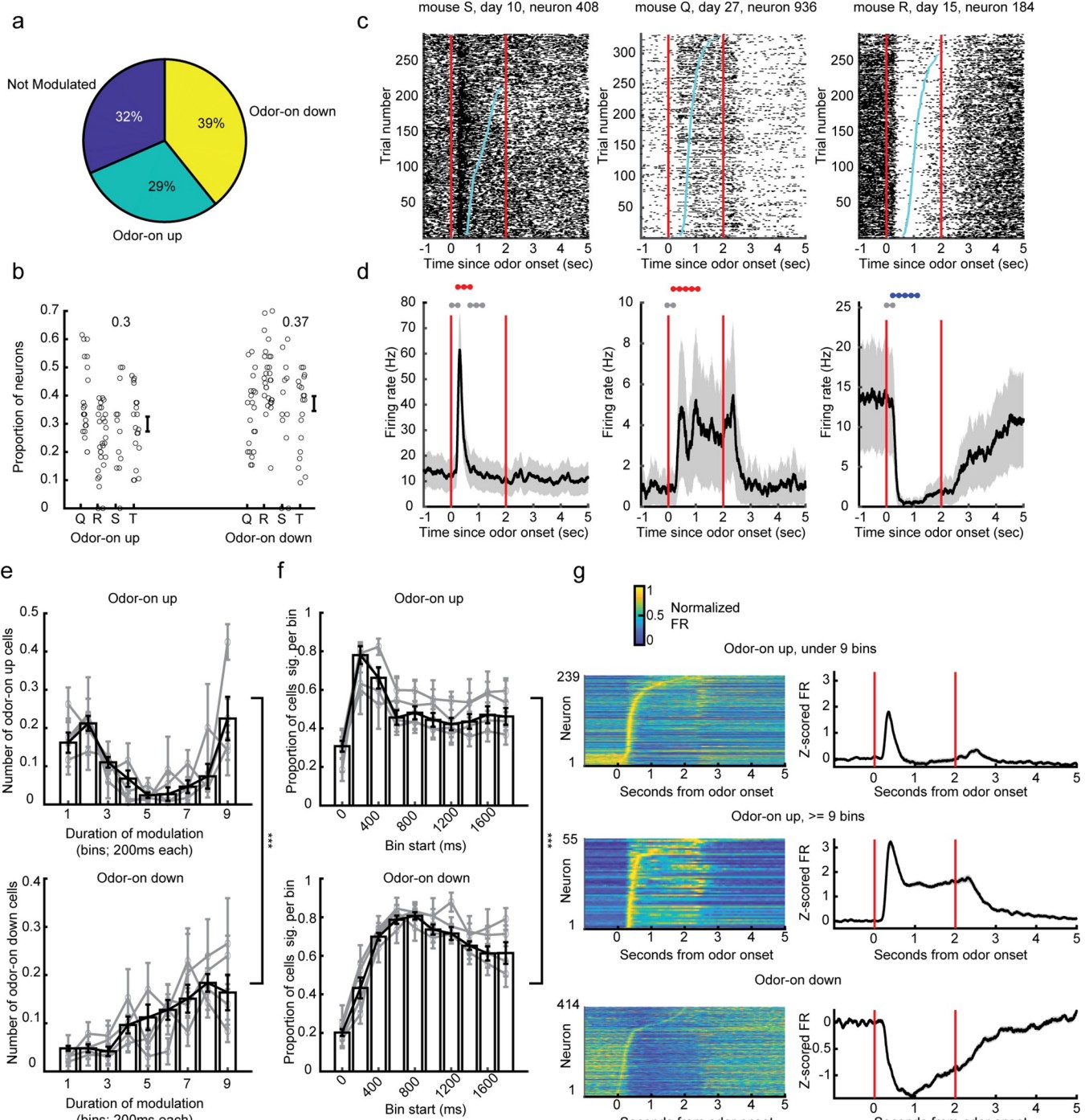

**Fig 2. Piriform neurons respond to odor onset.** (**a**) Proportion of all piriform neurons that significantly respond to odor onset by either increasing (odor-on up; green) or decreasing (odor-on down; yellow) their firing rate. (**b**) Proportion of neurons that significantly increase or decrease their firing rate across all sessions for all 4 mice (each small circle is 1 session). The mean proportion is in text above each condition. (**c**) Raster plots of 3 example neurons that respond positively (left two) and negatively (right) to odor onset. Red lines depict odor onset and offset; light blue dots depict lick times. (**d**) Peri-stimulus time histograms (PSTHs) of the same 3 example neurons as in (**c**). Time bins tested for significant modulation are marked above each trace (red is significantly positive; blue significantly negative). $P < 0.05$ adjusted for multiple comparisons. (**e, f**) The number of time bins in which neurons are significantly modulated (left) and the proportion of neurons that are significantly modulated in each time bin (right) are plotted for neurons that are significantly upwardly modulated at odor onset (odor-on up, panel **e**) and significantly downwardly modulated at odor onset (odor-on down, panel **f**). Gray lines represent the mean and SEM for each mouse separately. ANOVA details in Table 1, rows 5 and 6. (**g**) The normalized firing rate (left) and baseline-subtracted z-scored firing rates (right) for upwardly modulated neurons with fewer than nine 200 ms time bins significantly upwardly modulated (top), at least nine 200 ms time bins significantly upwardly modulated (middle) and downwardly

modulated neurons (bottom). Red lines depict odor onset and offset. All error bars and shading represent mean +/− SEM. *** $P < 1 \times 10^{-11}$. The underlying data for this figure are available for download from https://datadryad.org/stash/share/bC3NdXWDllJZYrRtq60q0WDQYhjZZuLulv91dm9WcYU.

odor period (Fig 2F and 2G, bottom). Whether we consider the duration of each neuron's modulation (Fig 2E; Table 1, row 5) or the proportion of neurons modulated per time bin (Fig 2F; Table 1, row 6), the odor-on up neurons tend to respond earlier and more briefly than odor-on down neurons.

Taken together, these data show that most piriform neurons are modulated at odor onset and that the direction and time course of their modulation are related.

## Piriform neurons respond differently to target and nontarget odor mixtures

We observed many neurons that have a higher firing rate for nontarget odor (Fig 3A; nontarget preferring) as well as those that have a higher firing rate during the target odor (Fig 3B; target preferring). There is a significant portion of odor-on upwardly modulated and downwardly modulated neurons that significantly differentiate between target and nontarget odor mixtures, but odor-on up neurons constitute a significantly greater proportion (Fig 4A and 4B; Table 1, rows 7 and 8). To test how unique this large proportion of target selective neurons was, we compared it to the proportion of neurons that were selective for probe or nontarget repeat odor mixtures. Nontarget repeats, which are randomly chosen from mixture of nontarget odors that mice do not need to differentiate from other nontargets, are particularly useful as comparisons to the learning of specific target mixture. We found that the target/nontarget discrimination is represented by significantly more neurons than either probe or nontarget repeat trial types (Fig 4C; Table 1, rows 9 and 10). Note that our analysis involves variable numbers of sessions (Target: 80, Probe: 21, NT repeat: 59) and neurons per session (Target: range = 3 to 30, mean = 13.2, median = 13; Probe: range = 5 to 26, mean = 12.1, median = 9; NT repeat: range = 3 to 20, mean = 13.5, median = 13). Because of these differences and differences in the number of trials, we also performed a down-sampling procedure to match between conditions and made a paired comparison across sessions and still found significantly more target/nontarget discriminating neurons than other types (S6 Fig). We explored the dynamics of the target-preferring (Fig 4D–4F, left) to the nontarget preferring (Fig 4D–4F, right) neurons and noticed that the target preferring neurons tend to be modulated early after odor onset (especially 200 to 600 ms bins), whereas the nontarget preferring neurons have a greater likelihood of being modulated in later time bins as well (Fig 4F; Table 1, row 11).

These data show that a substantial number of piriform neurons discriminate between target and nontarget odor mixtures, more than other salient odor mixtures, and that the time course of a neuron's response is related to its odor preference.

## Overtraining is characterized by latent improvements in piriform coding

After mice have learned the task, they exhibit consistent performance. Their accuracy as assessed by the choice index does not change across days (Fig 5A; Table 1, row 12). Additionally, the average lick latency does not change (Fig 5B; Table 1, row 13). We next wanted to see whether there was any covert learning that was not reflected in the mice's behavior. We turned to see whether the responses of piriform neurons were changing during this same period.

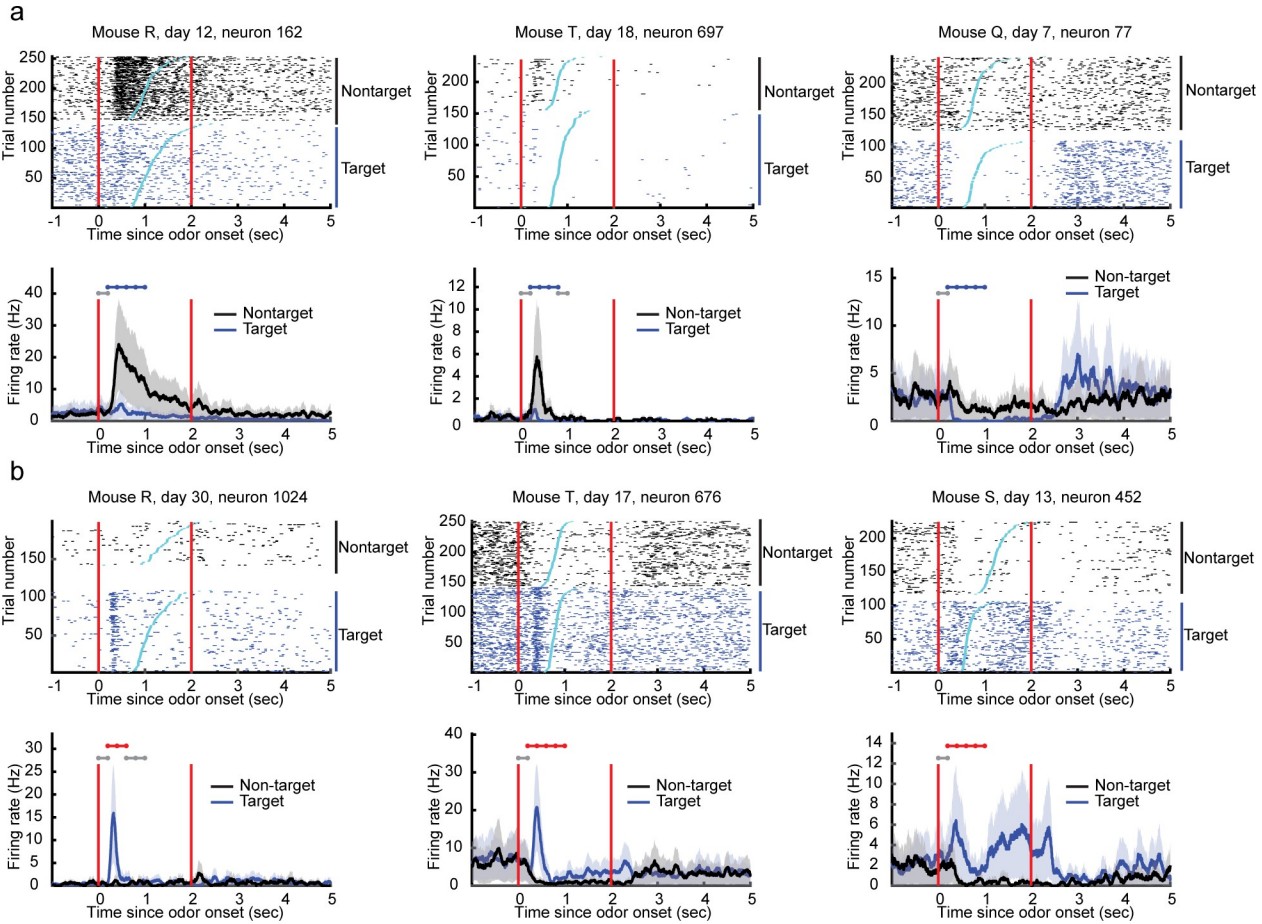

**Fig 3. Some piriform neurons differentially modulate their firing rate to the target and nontarget odor mixtures. (a)** Raster plots (top) and PSTHs (bottom) for 3 example neurons that respond significantly more to nontarget odor mixtures. **(b)** Similar plots for 3 other example neurons that respond significantly more to the target odor mixture. The underlying data for this figure are available for download from https://datadryad.org/stash/share/bC3NdXWDllJZYrRtq60q0WDQYhjZZuLulv91dm9WcYU.

Our experimental design included 2 nontarget odors that were repeated in each session (Fig 1B), which allowed us to interrogate how piriform cortex activity to these repeated nontargets evolved in the overtraining period. First, there are piriform neurons selective for a particular nontarget repeat, but they represented a smaller proportion than target-selective neurons (Figs 4C and 5C). The proportion of neurons on each day that were significantly selective for nontarget repeats (compared to all other nontarget odor mixtures; see Materials and methods) increases across days (Fig 5D; Table 1, row 14). Further, we found that the proportion of target selective neurons also increases across days (Fig 5E; Table 1, row 15).

To assess the possible consequences of this increase in the proportion of task-selective neurons, we performed categorical (target versus everything else) decoding using the population of simultaneously recorded piriform neurons on each day. We performed 10-fold linear discriminant analysis (see Materials and methods) decoding using the activity during the post-odor-on period of piriform neurons simultaneously recorded. We then compared this decoding accuracy to a shuffled distribution made by shuffling the labels of the category decision. We could readily decode category information for target, nontarget, and nontarget repeat trials (Fig 6A). Furthermore, although we were able to decode incorrect trials (average accuracy 60.0%, $N = 1,352$, Monte-Carlo significance test $P < 0.001$), decoding was more accurate when

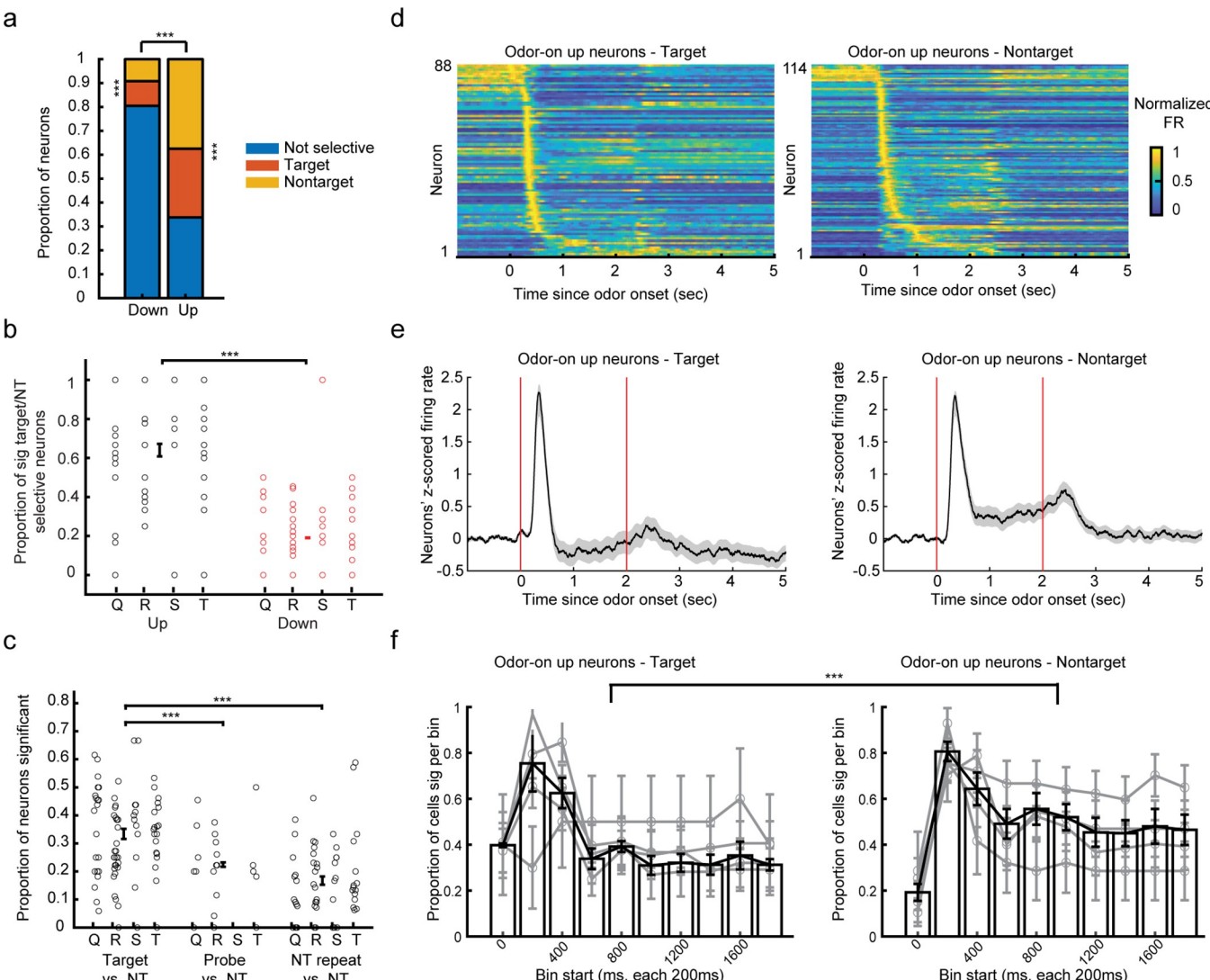

**Fig 4. Piriform neurons respond differently to target and nontarget odors.** (**a**) Proportion of all downwardly modulated neurons (left) and upwardly modulated neurons (right) that are significantly selective for target or nontarget odors. Target/nontarget label here refers to which condition the neuron fired more to. (Example nontarget and target neurons shown in Fig 3A and 3B, respectively.) Binomial probability of up or down neurons being selective for target/nontarget is $p < 7 \times 10^{-7}$ for both categories. A higher proportion of up-modulated neurons are selective for target/nontarget than down-modulated neurons (chi-squared test $p < 8.3 \times 10^{-35}$). Statistics details in Table 1, row 7. (**b**) Proportion of all upwardly modulated neurons (left) and downwardly modulated neurons (right) that significantly prefer target or nontarget odors across sessions of all 4 mice. Three-way ANOVA with direction (up or down), mouse, and session, details in Table 1, row 8. (**c**) The proportion of neurons that are significantly selective for target, probe, and nontarget repeats compared to nontarget odors across all sessions of all mice. Each small circle depicts a session. Three-way ANOVA with trial type, mouse, and session details in Table 1, row 9. Post hoc Bonferroni comparison details in Table 1, row 10. (**d**) The normalized firing rate of upwardly modulated neurons that respond significantly more to target odors (left) and nontarget odors (right). (**e**) The mean baseline subtracted z-scored firing rates of upwardly modulated neurons that respond significantly more to target odors (left) and nontarget odors (right). Red lines depict odor onset and offset. (**f**) The proportion of upwardly modulated neurons that respond significantly more to target (left) or nontarget odors (right) that are significantly modulated in each 200 ms bin. Gray lines represent the mean and SEM for each mouse separately. ANOVA details in Table 1, row 11. All error bars and shading represent mean +/− SEM. *** $p < 3 \times 10^{-4}$. The underlying data for this figure are available for download from https://datadryad.org/stash/share/bC3NdXWDllJZYrRtq60q0WDQYhjZZuLulv91dm9WcYU.

the animal made the correct choice (Fig 6B insert; Table 1, row 16), and decoding was more accurate during sessions when the mouse performed better (Fig 6B). This is what we would expect if the mouse were using piriform cortex information to make its decision. We predicted that we would see an improvement in decoding across days, as the mice gain more experience, even though they were already at ceiling levels of performance. We found that the decoder

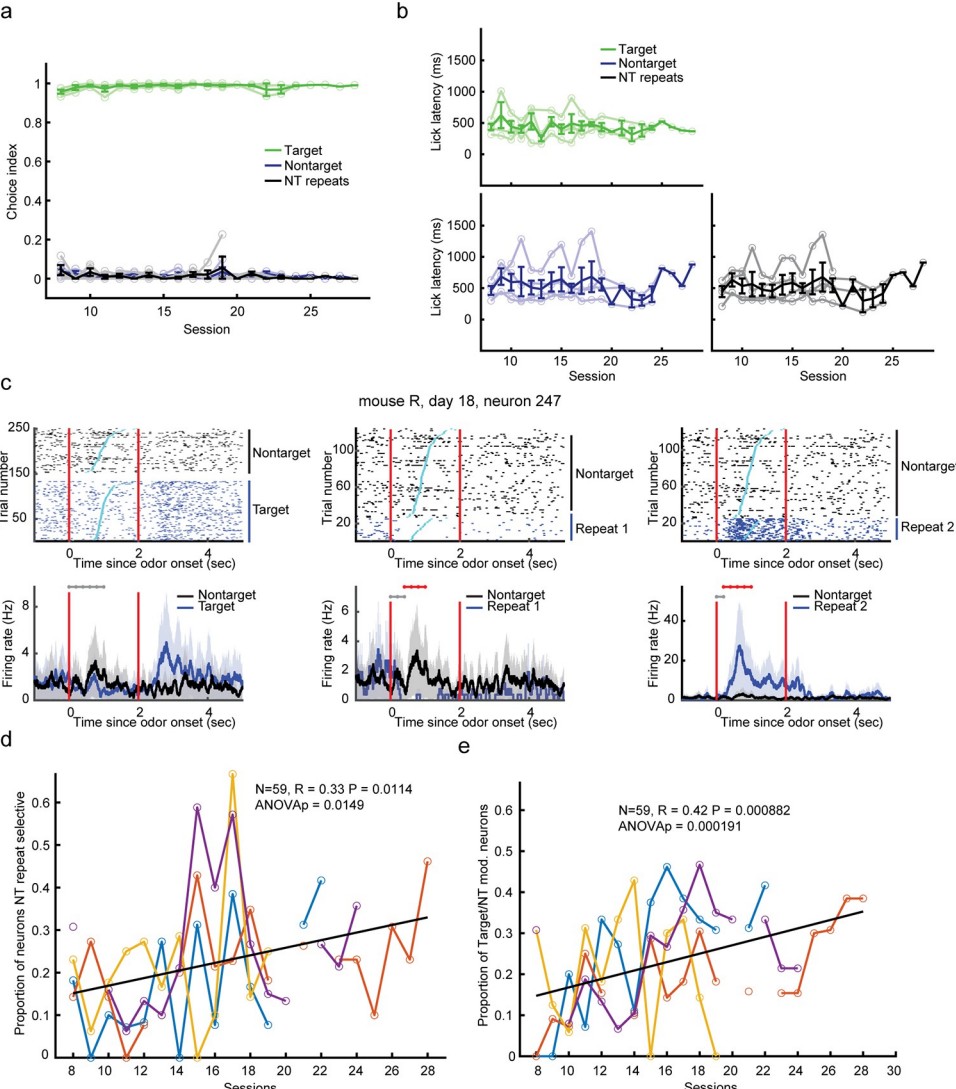

**Fig 5. Target/nontarget task with nontarget repeats.** (**a**) Choice index (1 for lick right, and 0 for lick left) for target, nontarget, and the nontarget repeat trial types across overtraining sessions. No change in accuracy across sessions (ANOVA in Table 1, row 12). Dark colors are averaged across mice; light are individual mice. (**b**) Lick latencies (ms since odor-on) across sessions for each trial type. No change in latency across sessions (ANOVA in Table 1, row 13). Dark colors are averaged across mice; light are individual mice. (**c**) Raster plot (top) and PSTH (bottom) for an example neuron's response to target compared to nontarget odors (left) as well as to 2 nontarget repeat trial types (middle and right). Note different y-axis scale in bottom panels. (**d**) Proportion of neurons selective for either of the 1–2 nontarget repeat trial types vs. nontarget trials across days. Black is a best-fit line. Each color is a different mouse. Significant change across sessions (ANOVA in Table 1, row 14). Note that breaks the data represent days that the mouse was trained but neural data were not available. (**e**) Proportion of neurons significantly selective between target/nontarget odors across days. Black is a best-fit line. Each color is a different mouse. Significant change across sessions (ANOVA in Table 1, row 15). All error bars and shading represent mean +/− SEM. The underlying data for this figure are available for download from https://datadryad.org/stash/share/bC3NdXWDllJZYrRtq60q0WDQYhjZZuLulv91dm9WcYU.

performance during correct trials increases across days for target, nontarget, and nontarget repeat trials (Fig 6C; Table 1, row 17–19). We also performed a boot-strap analysis to verify that this increase was significantly greater than expected given our data ($p < 0.02$), by shuffling the day of each session to generate a null distribution and found that the true slopes exceed this null distribution for each trial type (Fig 6D).

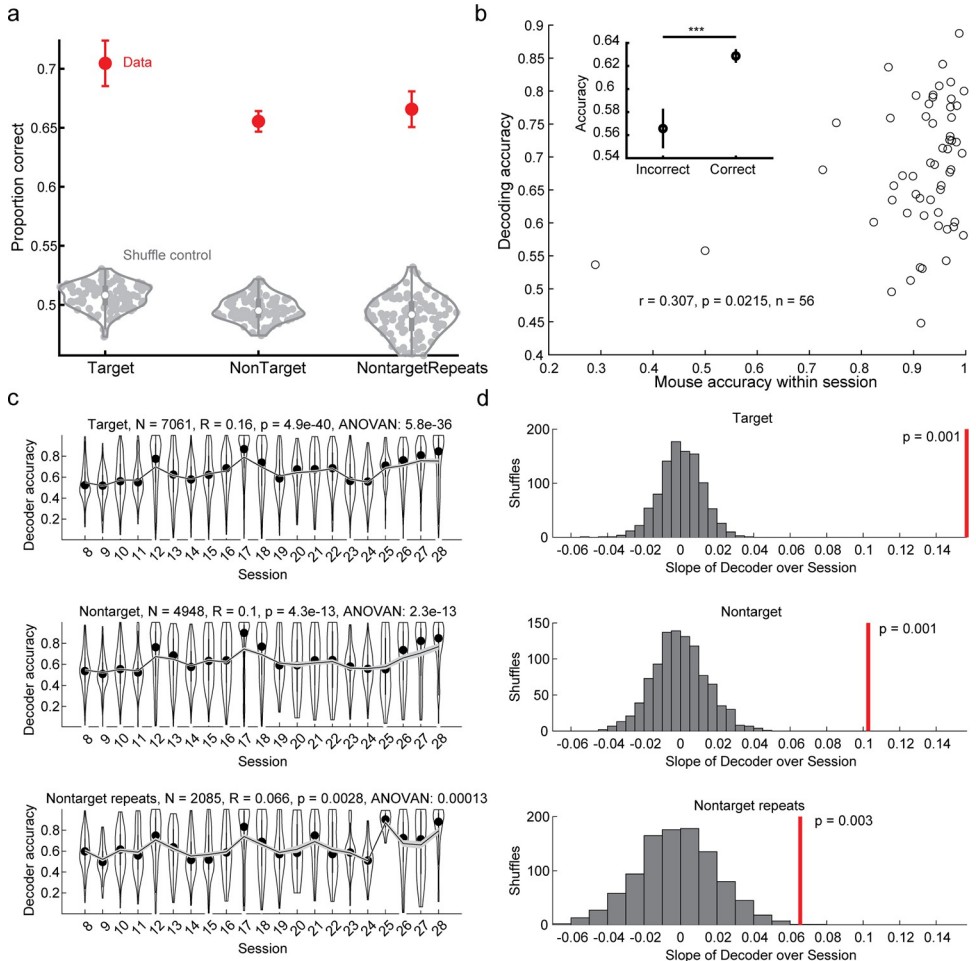

**Fig 6. Decoding in the piriform population improves across sessions during the target/nontarget discrimination task.** (**a**) Accuracy of decoding the categorical choice of the 3 trial types (red) compared to a shuffled distribution (gray clusters). All trials were averaged within each mouse. Error bars represent mean +/− SEM across mice. Monte-Carlo significance test all $P < 0.001$. (**b**) The mean accuracy of category decoding plotted against the mouse's average behavioral accuracy across all sessions. The Pearson's correlation is shown. Inset: the mean accuracy of the category decoder for incorrect compared to correct trials after controlling for factors of repeat number, trial, session, and mouse. ANOVA details in Table 1, row 16. *** $p < 0.005$. (**c**) The mean decoder accuracy across days during correct trials for each trial type. Data are shown as kernel density estimates. The continuous gray lines indicate the mean values, and the black dots indicate medians. Top: target, Middle: nontarget, and Bottom: nontarget repeat. The relationship between session and accuracy assessed with a Pearson's correlation (also significant in an ANOVA, details in Table 1, rows 17–19). (**d**) Pearson's correlation calculated in c compared to a distribution of correlations calculated with shuffled (1,000 times) session numbers. $P$ values from Monte-Carlo significance test. The underlying data for this figure are available for download from https://datadryad.org/stash/share/bC3NdXWDllJZYrRtq60q0WDQYhjZZuLulv91dm9WcYU.

Taken together, these data show that during the overtraining period, while mice do not alter their behavior significantly, the underlying piriform population representation is changing. More piriform neurons are being recruited to be target and nontarget repeat selective, and the discrimination of the 2 categories improves within the piriform population.

## The dissimilarity across category coding predicts accuracy on probe trials

Although the increase in task-selective neurons and the improvement in population decoding does not benefit the mice by improving their performance during overtraining, we

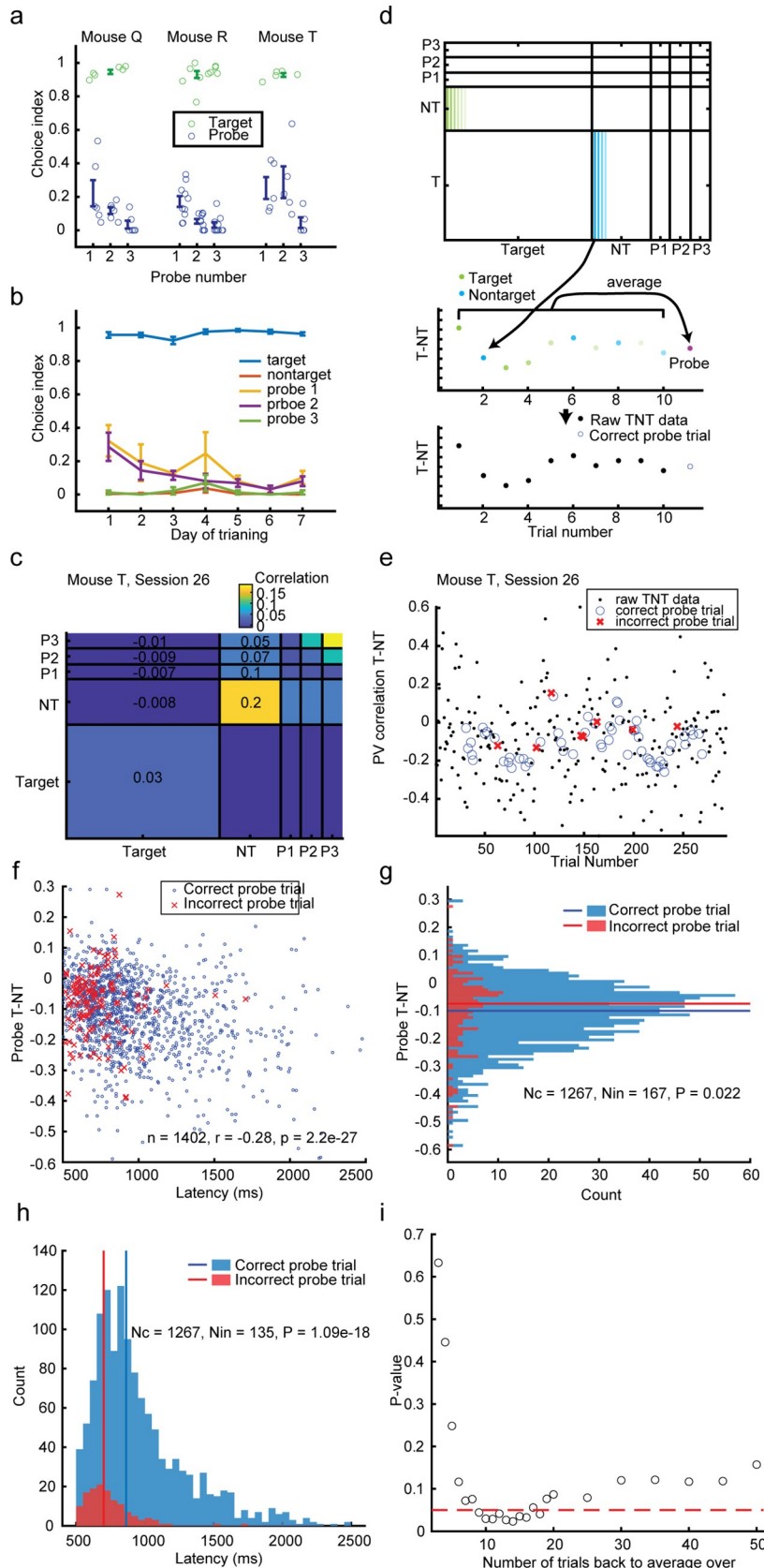

**Fig 7. Dissimilar target/nontarget population responses relate to better subsequent performance on probe trials.**
(**a**) Choice index of 3 probe types across 3 mice. Each small circle is 1 session. (**b**) Choice index over days of training with the probe trials. Probe trials improve across sessions (see Table 1, row 20). (**c**) An example of population vector correlations (the Pearson's correlation of the activity of piriform neurons in 2 different trials) across trials in 1 session, averaged within trial-type categories (P1–3 are probes 1–3). (**d**) Schematic of the procedure of determining the target-nontarget similarity (T-NT) just preceding the probe trials (probe T-NT). Top: The correlation between each target trial and all other nontarget trials was averaged to produce a T-NT for target trials (green), and vice versa for nontarget trials (blue). Middle: For probe trials (pink), the similarity is plotted as the average value of the 10 preceding trials, as a surrogate of the state of the piriform cortex when the probe trial arrives. The pink dot (probe trial) is the average of the blue and green dots (T-NTs for 10 preceding target and nontarget trials). Bottom: The example trial (here the mouse responded correctly) coded in the same colors as in **e**, which is an example session. (**e**) The T-NT for target, nontarget, and probe trials from the same example session as in **c** using the procedure described in **d**. The probe trial similarities are colored according to the mouse's behavioral response—correctly classified as nontarget (blue circles) or incorrected classified as target (red crosses). (**f**) The immediately preceding T-NT similarity for each probe trial plotted against lick latency for the probe trial across all mice (blue for probe trials with correct responses and red asters for incorrect responses). Negative TNT values represent a more differentiated piriform representation. There is a significant negative correlation (Pearson's correlation $R = -0.28$, $N = 1,402$, $P = 2.2 \times 10^{-27}$). Monte-Carlo significance test $p$-value = 0.001. (**g**) The immediately preceding T-NT similarity of each correct (blue) and incorrect (red) probe trial. Correct trials have significantly less T-NT similarity than incorrect trials (one-sided signed-rank test, $P = 0.022$; Monte-Carlo significance test $p$-value = 0.013). (**h**) The lick latency of each correct (blue) and incorrect (red) probe trial. Correct trials have significantly higher latencies than incorrect trials (one-sided signed-rank test, $P = 1.09 \times 10^{-18}$). (**i**) The same analysis computed in g but with a $p$-value computed across different look-back variables. Red dotted line depicts $p = 0.05$. Error bars represent mean +/− SEM. The underlying data for this figure are available for download from https://datadryad. org/stash/share/bC3NdXWDllJZYrRtq60q0WDQYhjZZuLulv91dm9WcYU.

hypothesized that building more discriminable categories would benefit the mice when presented with more difficult decisions (Fig 7). After they were experts at the categorization task, 3 mice then moved on to sessions that included a new trial type, the probe trial. Probe trials consist of one of the 3 odors in the target mixture and 2 randomly chosen odors from the nontarget odor pool (Fig 1A). Mice were able to perform well across all 3 types of probe trials (Fig 7A), and they improved across days of training with these trial types (Figs 1G and 7B; Table 1, row 20).

We used pairwise similarity analysis to assess how strongly the piriform population separates target and nontarget representations. The normalized Pearson's correlation coefficient of the activity of simultaneously recorded neurons was taken for each trial (Fig 7C and top of Figs 7D and S7; see Materials and methods: Population vector correlations). For each target or nontarget trial, we calculated the "target/nontarget similarity," which is the average population vector correlation between that trial (e.g., a target trial), and all of the opposite (e.g., nontarget trials) category's population vectors (Fig 7D, each blue and green dot represents the "target/ nontarget similarity" for that trial and comes from averaging the blue and green lines). We hypothesized that the state of the target/nontarget representation during each probe trial could predict how the mouse would perform on that probe trial. To estimate the state of the cortex on each probe trial (pink dot in Fig 7D), we averaged the target/nontarget similarity in the 10 trials preceding the probe trial and plotted this value for each probe trial (Fig 7D and 7E; averaging the blue and green dots to get the pink dot in **d** and in **e**, averaging the black dots to get the blue and red markers). This produced an extrapolated value of the target/nontarget similarity during probe trials based off the running average of the last 10 trials.

Interestingly, probe trials with correct responses tended to have longer lick latencies than probe trials with incorrect responses (Fig 7H). This was also true for target and nontarget trials, but the speed accuracy trade-off was especially stark in probe trials (S8 Fig; also see S9 Fig for more information on lick latencies).

Therefore, we looked at how this latency relates to the state of the piriform cortex. Across all sessions and mice, we found that the lick latency in a probe trial was inversely correlated with the target/nontarget representation similarity preceding a probe trial (Fig 7F; Pearson's

correlation R = −0.28, $N$ = 1,402, $P$ = $2.2 \times 10^{-27}$). Negative similarity values represent a more differentiated piriform representation, and this differentiation tends to happen on trials that mice take more time to respond to. Furthermore, by looking at this same "target/nontarget similarity" readout for probe trial types by whether they were correct (blue) or incorrect (red), we found that correct probe trials tended to have more dissimilar representations between target and nontarget in the preceding trials (Fig 7G). We then explored the parameter space of the number of trials we looked-back to interpolate the target/nontarget similarity and found that this relationship with accuracy is consistent and significant across look-back sizes of 9 to 18 trials (Fig 7I).

Therefore, although the improvement of the category representation in piriform during overtraining did not change behavior at the time, it was subsequently related to better performance in more challenging probe trials.

## Discussion

We recorded from pPC neurons as mice performed a categorization task to classify 1 target mixture of 3 odors as different from hundreds of other 3-odor mixtures. Our main findings are 1) that the target odor mixture becomes overrepresented with more pPC neurons distinguishing the target odor mixture than the other odors repeated in the task 2), that even without any stark benefit in behavior, the pPC representation becomes more robust and selective with overtraining, and 3) that this selectivity benefits behavior during a later perturbation with more difficult and ambiguous probe trails.

### Task and behavior

We developed a novel task in which mice need to distinguish 1 unique odor mixture from hundreds of odor mixtures and found that mice learned this task rapidly. We saw a slight dip in performance when we introduced probe trials, with mixtures that include one of the odor components from the target odor mixture along with two others. However, this performance dip was once again followed by rapid learning to distinguish the target odor mixture, specifically.

In our experiments, the latency from odor onset to when mice licked was approximately 500 ms, of which approximately 100 ms was accounted for by the time taken for the odor to reach the mouse's nose from the last solenoid valve (see Materials and methods). Previous studies have shown the latency depends on the difficulty of the olfactory task [19,20]. While these earlier studies reported faster reaction times, a recent study [18] in which mice were trained to report the presence of 1 target odor within a background of other many other odors, reported reaction times similar to those in the present study (approximately 500 ms). The behavioral setup may be important for setting limits on an animal's reaction time. Shorter reaction times have been reported in task setups different from ours [20–22], but reaction times were similar to ours in studies with similar behavioral setups as ours, with head restraints and choice reporting using left versus right licks [18,23]. With similar behavioral setups, longer reaction times were also reported in nonolfactory tasks [24,25].

### Overrepresentation of relevant odors

A few types of learning occur that are reflected in an overrepresentation in individual piriform neurons' selectivity. The largest overrepresentation was of the target odor mixture, likely showing the effect of both the repeated exposure of the odor mixture and the reinforced dimension of the task (mice are rewarded for categorizing target and nontarget odor mixtures by licking to 2 different lick ports). However, a significant proportion of pPC neurons are also selective

for odor mixtures that mice were not forced to distinguish from others. A significant proportion of neurons differentiate probe trials, where only one of the 3 odors matches the target mixture, from nontarget mixtures, even while mice are rewarded for classifying these into the same category. Further, a significant proportion of neurons differentiate nontarget repeats (which are repeated more often in any given session), compared with other nontarget mixtures. This is the case even though mice are rewarded for classifying these trial types the same way as other nontarget mixtures. This proportion of neurons selective for the nontarget repeat also increases across sessions. These results are consistent with the interpretation in that there is a strong associative encoding within the piriform but also still strong pure sensory encoding as well. Instead of conforming completely to the task and collapsing all nontarget mixtures together, the piriform still can distinguish differences between these odor mixtures.

Although piriform cortex can still represent individual odor mixtures, its robust target coding means it already has all relevant information that can be used by a downstream area to compute a categorical decision. Our data do not speak to whether this learned overrepresentation is bottom-up, top-down, or produced from dynamics within piriform. However, it is likely that, while the afferent inputs are relatively set, the intracortical inputs are important for this learning [26–29].

This study only considers pPC neurons, and there is reason to believe that aPC neurons could behave differently. There are anatomical and structural differences between anterior and posterior piriform [30–32]. These are accompanied by differences in encoding, with pPC being described as more associative and less sensory-driven than aPC [16,33].

Our data also do not speak to how much of the overrepresentation of target-selective neurons occurs from repeated experience or from learning. We do not have neural data from early learning to see how this overrepresentation emerges, and the target condition is the most frequent as well as the most reinforced. Both experience [34] and reinforcement [16] have led to overrepresentation of selective neurons within the piriform cortex, and we believe both forces are at play here. Schoonover and colleagues also found that experience alone can also reduce drift of the aPC neural representation over time [35]. Exploration of that effect cannot be assessed here, as we did not follow the same neurons over days. It is likely that the active component and the go-go structure of our task both contribute to differences in pPC representation along with the mere exposure of odors [36], and future work exploring how each of these task demands change separately across time would be fruitful.

### Profile of target-selective neurons' activity

Our finding that target/nontarget-selective neurons are more likely to be upwardly modulated is consistent with findings that inhibition is more broadly tuned than excitation [3,6,12,37,38]. In fact, the combination of fast upwardly modulated odor specific neurons and slower inhibited neurons could be indicative of circuit dynamics affecting population selectivity within the piriform [29,39,40].

Further, we found that target-preferring neurons are more likely to increase their firing rates quickly after odor onset than nontarget-preferring neurons. Because there are hundreds of different combinations of nontarget mixtures, compared to the fixed 3 odors in the target mixture, these nontarget responses are likely drawing from a wider range of latencies earlier in odor processing [41–43] and subsequently also have a wider distribution of feedback dynamics due solely to the huge distribution of odor mixtures it is based on. However, some of this effect could be due to learning as well, since recognizing the target odor mixture (e.g., by template matching) may be the fastest method of making an accurate decision using just identity coding. Perhaps this temporal refinement and truncation is due to recurrent circuitry within the pPC itself [29,44].

## Other notable neural correlates

We note that a number of lick-sensitive and odor-off sensitive neurons also exist in the piriform cortex. Lick correlates could arise from motor or taste responses [45]. Although these were not the focus of our investigation, we reported the prevalence and were able to remove those neurons to concentrate on odor responses, and to assure ourselves that task decoding was not dependent on these responses. Odor-off responses have been reported in other species in olfactory receptor neurons and antennal lobe [46–48], as well as in mouse aPC [49]. We expect these dynamics are likely due to feedback from within or outside the piriform circuit and not driven from the bulb, as we did not find a significant number of neurons that showed odor-off responses without any odor-on response.

## Neural changes despite behavioral stability

During a period of overtraining, when mice were already at ceiling in performance, we saw 2 types of changes in the pPC population. A greater proportion of neurons become target-selective, making the target coding an even more robust population response, reflected also in the increase in category decoding during this period as well. In addition, a greater proportion of neurons become selective for nontarget repeats, even as the task incentivized generalizing across all nontarget mixtures instead. This learning was purely due to exposure and did not sacrifice population decoding of the target/nontarget category decision, since the population decoding accuracy actually increases during that time for those nontarget repeat trials. Thus, when mice were experts and overtrained on this odor discrimination task, latent learning of aspects of the task not currently relevant may have occurred without sacrificing task accuracy [50]. Improved discrimination of nontarget repeats from all other nontargets occurred through trials where the mouse was rewarded, but not to improve that discrimination. Thus, it could also be considered perceptual learning, perhaps occurring orthogonal to the reinforced dimension of target versus nontarget [51]. This could be due to attention or the coincidence of reinforcement signals, which arise to strengthen the task-relevant dimension, which is also theorized to support task-irrelevant perceptual learning in vision [52].

The nontarget repeat mixture was also novel each day, meaning that this learning is a type of rule learning different from stimulus-specific sensory learning [53–55]. Another implication of this finding is that as demonstrated with a simple decoder, any area downstream of the pPC can already have strong category information, which could immediately be acted on without a second level of processing. This could mean that the mouse may be more reliant on the piriform than higher level areas such as the OFC during this type of task [38]. Future experiments are required to know whether and when the piriform cortex and other areas are necessary for this task.

One caveat of this effect is that as recording sessions progressed, tetrodes were moved slightly by the experimenter each day to attempt to record from a new population of neurons. Although we cannot rule out layer differences, we do not think it is the most likely explanation for the effect. The overall trend of this movement was likely to be downwards (moving more ventrally), so earlier sessions of overtraining would be sampling deeper layers and later sessions sampling more superficial layers. As layer 2 is denser than the deeper layer 3 or more superficial layer 1, we calculated our neuron yield across these days as a proxy for density. We found that the yield decreases across days, perhaps due to a move from layer 2 towards 1 (2b to 2a), which would correspond to increasing proportion of semilunar cells. However, prior work has shown that semilunar cells are less selective for odors [56]. Thus, this hypothetical is not consistent with the current knowledge of how layers and selectivity interact. Although we cannot fully rule this possibility out, we believe that due to this reasoning, along with the fact that

tetrode movements were not large and can often lead to nonlinear movements, the most reasonable explanation for the changing selectivity is still the continuous experience the mice accumulate.

## Importance of training history

Our data are consistent with prior work showing that rewarding animals for separation of odor types will lead to separation in piriform representation [57,58], but our data extend this from looking only within a mixture of odors to across different mixtures of odors. Interestingly, while Chapuis and Wilson's experiments showed that experience reinforcing pattern separation leads to better pattern separation in pPC and experience reinforcing generalization leads to better generalization, we saw a slightly different pattern. Our task reinforced generalization across all the nontarget odors used for nontarget odor mixtures, and yet the pPC still showed pattern separation within that category, with nontarget repeats becoming more selective across experience.

The slight differences in population dynamics within these data highlight how important the details of an animal's training history are to its cortical representation. In experiments in the olfactory bulb, training mice on easy or difficult odor discrimination leads to changes in robustness and efficiency in olfactory blub coding [59]. The overrepresentation of the target odor mixture is a less efficient, but more robust, coding scheme. It then becomes more and more robust with experience. One could imagine that with different protocol of training, a more efficient and less robust code may be produced instead. In fact, in other behavioral setups, pPC neurons have been shown to become less selective, making them better at categorization than aPC [60].

Training history has also been known to affect cognitive flexibility, reflected in the "overtraining reversal effect" [61], whereby the overtraining of a task results in better performance during a subsequent change in the task. These studies predominantly concern spatial and visual tasks, but effects have also been seen in odor-guided tasks [62].

Although the changes in single neuron representation during the period of overtraining and the resulting increases in population decoding accuracy do not benefit the mice at the time, we hypothesized that they may benefit them when the task becomes more difficult. We found that in periods when overtrained mice had better separation between target and nontarget representations, they gave probe trials more time before responding and were more accurate in their responses. This perturbation highlights the utility of piriform building a robust representation of the task. In an uncertain world where task difficulty and goals can change, the piriform is able to prepare for possible future reward without sacrificing current reward rate.

## Materials and methods

### Surgery

C57BL/6J Mice from Jackson Laboratories, all male, 8 to 10 weeks old of age at start of the experiment. Experiments were conducted in accordance with Harvard University Animal Care Guidelines and the "Guide for the Care and Use of Laboratory Animals" (National Research Council, eight edition, 2011). Protocol 29-20-4 was approved by Harvard Faculty of Arts and Sciences' Institutional Animal Care and Use Committee. Surgeries were performed on naive animals, and all behavioral training began after recovery from surgery. Mice were anesthetized (Ketamine/Xylazine, 100 and 10 mg/kg, respectively). Following surgical implantation of a tetrode drive and a custom head plate, all mice we rehoused individually. A cranial window (approximately 1 mm) was made over the dorsal skull at a location directly above the

area targeted for electrophysiological recordings with the goal of implanting a tetrode bundle (8 tetrodes plus a 200-um-diameter optic fiber to ensure stability). Posterior piriform coordinates: 0.5 mm posterior and 3.8 mm lateral from bregma, and 3.8 mm ventral from brain surface. (All recordings occurred ipsilateral to the reward port associated with the target odor combination.) To ensure stability of the head of the animal during behavior and recording at a later stage, a custom-made head plate (made of lightweight titanium; dimensions, $30 \times 10 \times 1$ mm; weight, 0.8 g) was affixed to the skull. A shallow well was drilled over the posterior lateral skull, and a single skull screw was affixed at that location. A wire was attached to this skull for grounding electrophysiological recordings. In addition, a plastic cone was positioned around the tetrode drive, and capped with a removable lid, to prevent damage to the drive. Following the completion of the surgery, mice were given 1 week to recover before behavior training. Recordings were completed using openEphys (https://open-ephys.org/) for data acquisition and MClust (https://redishlab.umn.edu/mclust; https://github.com/adredish/MClust-Spike-Sorting-Toolbox) for spike sorting software.

## Behavior training

Mice were water restricted in compliance with approved protocols. Mice were acclimatized to the behavioral apparatus for 1 session in which they were allowed 30 minutes of free exploration with free water available. This was followed by an additional day in which they were head restrained and were allowed to lick for water from the 2 ports. In this session at any given time, only 1 water port could deliver water, and the mouse had to try both sides to discover which one. If the mouse collected the water drop from that port, the next water drop had a 50/50 chance to be in either port; if the mouse only picked one side to lick, it never received additional drops unless it tried the other port. Generous manual delivery of water drops to the "correct" side occurred to help the mouse learn that there were 2 water ports and to avoid a side bias.

In the next phase of behavior training, the trial structure was introduced, where the odor was delivered for 2 seconds and a free drop of water was available at the correct lick port (for target the right lick port and nontargets the left lick port), and a 10-second intertrial interval. This shaping phase of the behavior training lasted 1 to 2 sessions until the mouse refrained from licking during the intertrial interval.

The next training phase consisted of blocked trials—3 of target and 3 of nontargets; in these trials, the mouse had to lick the correct port to receive a water reward. When performance was greater than 70% correct for both target and nontarget trials, mice were moved to the next phase of training (typically 3 to 4 days of block sessions).

The next phase had no structure, but we forced trials on both lick ports. If the mouse started, e.g., with a target and licked correctly, the next trial would be target or nontarget with equal probability; if the trial was incorrect, the next one would still be a target. When performance was greater than 70% correct for both target and nontarget trials, mice were moved to the next phase of training (typically 2 days of no structure force both sides).

The next phase had a random trial structure where every trial had a 50% chance of being target or nontarget. In the initial training phases, if the mouse licked the incorrect port first but then made the correct lick during the decision period (odor on 2 seconds + 500 ms), water reward was released from the correct lick port; however, the trial was still counted as incorrect.

After performance was greater than 70% on both sides, an additional criterion was added wherein the lick choice had to be to the correct side only.

When behavior performance was greater than 90% for at least 3 days, the mouse was considered expert at the target/nontarget odor mixture task, and probe trials were introduced in subsequent sessions.

## Data analysis

All analysis was performed using custom scripts in MATLAB R2022a, which are available on GitHub at https://github.com/ABernersLee/TargetPaper_20220822.

## Analysis of behavior

Odor onset is demarcated as when the last solenoid switched on. There was a consistent delay between this time and when the odor reached the mouse, which was about a hundred milliseconds. We excluded the small number of trials (1.7%) in which the first lick came before 500 ms after odor onset (S8 Fig). Subsequent analyses, such as the lack of change in lick latency across overtraining, remain remarkably similar qualitatively and quantitatively when these short latency trials are instead included.

Choice index for target trials is the proportion of total trials that were correct, and for other trial types (nontarget trials, probes, nontarget repeats), it is 1 minus this value. The total number of trials was calculated in 2 different ways. In Fig 1, it was calculated during all trials, and trials when the mouse did not lick left or right to indicate a choice were deemed incorrect. Only trailing trials where the mouse did not lick were truncated from the analysis (the last trial where the mouse licked was deemed the last trial). S1 Fig was also calculated this way. In Fig 5, it was calculated only using trials in which the mouse made a decision, meaning that they licked left or right during the response period. The effects reported are stable between the 2 ways of calculating the choice index.

## Odor-on significance of individual neurons

We calculated the significance of firing rate changes due to odor presence or lick timing by comparing the firing rates during odor- or lick-related time periods in the task to the baseline firing rate (first 4 seconds of the trial and the last 4 seconds of the trial; 5 to 1 seconds before odor-on and 6 to 10 seconds after odor offset). To assess each time bin's significance, we calculated the modulation [(bin − baseline) / (bin + baseline)] and compared it to a distribution of jittered spike trains where each trial was circularly shifted independently by a random amount of time. We calculated a $p$-value as the percentile of the absolute value of the real modulation compared to 1,000 of those produced by a jitter procedure.

To identify significantly odor-modulated neurons, we found neurons that had any of the five 200-ms bins in the 1 second after odor-on with a $p$-value of less than 0.01 (0.05/number of bins tested). Neurons were defined to be odor-on upwardly modulated if any of those significant bins were positively modulated; neurons with only negatively modulated significant bins were considered odor-on negatively modulated. Only neurons that were not found to be lick-modulated (see next paragraph) were designated as odor-modulated.

To find any neuron that could be contaminated by lick modulation, we aligned each neuron's firing to either the odor (odor onset and odor offset) or the first lick (lick-on) and calculated the average peri-stimulus spike rate for both alignments in 200 ms bins. We evaluated the entire odor period (odor-on to odor-off; 2 seconds) and the lick-on adjacent period (1 second around the first lick; 400 ms before to 600 ms after). For comparing firing rates between lick-on and odor-on periods, we only used trials where the lick occurred at least 800 ms after the odor-on time and only considered bins that had a $p$-value of less than 0.05. For neurons that had a positive odor-on modulation bin, if any lick-on adjacent bin was significant and had a higher firing rate than the firing rate in the positive modulation and significant bin, then the neuron was deemed lick-contaminated. For neurons that did not have a positive significant modulation bin, if any lick-on adjacent bin was significant and had a lower firing rate, then that neuron was considered lick-contaminated.

### Depicting the time course of neurons' modulation

To show the responses of all the neurons in a given category (e.g., odor-on up or odor-on target preferring), we plotted a colormap of all the neurons across −1 to 5 seconds around odor onset. Each neuron's smoothed firing rate was normalized by dividing by the maximum firing rate during that period. Then, to show the population response, we z-scored each neuron's response during this same period and subtracted the average firing rate from the second before odor onset for each neuron.

### Significance of odor-identity modulation

To determine whether a neuron has a significantly different firing rate between 2 odor groups (target versus nontarget, nontarget repeats versus nontarget nonrepeats, or probe versus nontarget), the difference in the firing rate in the same odor-on bins was used to identify odor-on significance (0 to 1 second in five 200-ms bins) between the 2 groups of trials. The unsigned average difference between the 2 groups was compared to the unsigned differences in a shuffled distribution where the group labels were shuffled 1,000 times and the percentile of the real value compared to that distribution produced a $p$-value. A neuron was considered significantly selective for a group if any of the 5 bins had a $p$-value of less than 0.01 (0.05/number of bins tested).

### Odor-off responses

To determine whether a neuron had a significant odor-off response, we used a similar method to that of determining odor-on significance but with a few more necessities to exclude responses that were not specific to the odor-off period. Neurons needed to satisfy 3 requirements: (1) Neurons needed to have at least 1 bin in the second after odor offset with a $p$-value less than 0.01 (0.05/5 bins) compared with baseline. (2) Neurons needed to have a significant modulation (in the odor-off period) from the odor-on period. This was evaluated by taking the modulation in the 500 ms at the end of the odor-on period (1.5 to 2 seconds after odor-on) and comparing it to the 100 to 600 ms after odor offset. The modulation [(pre − post) / (pre + post)] was compared to a distribution of 1,000 times that the trials were independently time-jittered and the same modulation was calculated. Neurons needed an unsigned modulation exceeding the 95th percentile of the jitter distribution ($p$-value of 0.05). (3) Neurons needed to have the same direction of modulation from both comparisons; e.g., if a neuron had a positive modulation compared to baseline periods, but a negative modulation compared to odor-on periods, this neuron would not be considered to have an odor-off response.

### Trial down-sampled comparisons

To check the validity of comparisons between proportions of target or probe/repeat selective neurons, which have different numbers of trials, we used a down-sampling procedure. Here, a reduced number of trials, two-thirds of the number of repeat or probe trials in a session (the trial-type with the least number of trials, thus equating the number of trials between the 2 types), was randomly chosen from the trial type (target, probe or repeat), and these trials were used to compare against the nontarget trials in that session in the same procedure as outlined above in Significance of odor-identity modulation. This procedure was performed 50 times for each neuron in each session, for each trial type. The mode of these 50 iterations' $p$-values for each neuron was used to calculate the proportion of neurons that passed significance, and these proportions were compared with a one-sided signed rank test.

### Changes during overtraining

To calculate the change in the proportion of selective neurons across days, we looked at 2 types of neurons. We looked at the proportion of nontarget repeat selective neurons and the number of target/nontarget selective neurons. We calculated a Pearson's correlation and ANOVAs to determine changes in proportions across sessions.

### Population analysis

This section describes analyses applicable for both the decoding and population vector analyses. For each of the 2 experiments (probe and nontarget repeat), each session where this stimulus was included was analyzed separately (this includes one session where both experiments took place). For each simultaneously recorded set of neurons, the post-odor onset time window of 100 ms to 600 ms was taken for each neuron. We excluded neurons that fired fewer than 5 spikes summed across all trials in the post-odor onset time window. We excluded sessions with fewer than 4 neurons that passed this criterion. The trial by neuron matrix was z-scored for each neuron such that each neuron contributed equally to subsequent analysis.

### Decoding during overtraining

We performed 10-fold decoding (linear discriminant analysis) by randomly separating 90% of the trials into training data and 10% into test data. We did this 10 times and took the average of these 10 iterations. We used the classify.m function to perform linear discriminant analysis on the category decision (i.e., target versus everything else). To assess significance of the accuracy of this decoding, we performed 1,000 shuffles whereby the category decisions were shuffled and the same procedure as described above was performed. We then performed a Monte-Carlo significance test to assess whether the accuracy of our decoding exceeded chance levels. To analyze changes across conditions (when the mouse was correct, or across days), we used the estimate of the posterior probability (using the third output of classify.m function) that the decoder chose the correct trial type as a more fine-grained measure of accuracy.

To analyze changes across the mouse's accuracy, we performed an ANOVA with factors: repeat number, trial, day, mouse, and mouse's accuracy. Both trial and day were continuous factors. We verified that there was a significant main effect of mouse's accuracy in that ANOVA and reported the *p*-value. We then used multcompare.m to assess the direction of the difference. We reported the Tukey's post hoc test *p*-value and plotted the estimates and standard errors of each group.

To analyze changes across days, for each trial type separately (e.g., target), we performed an ANOVA with factors: repeat (when applicable), trial, day, mouse, and mouse's accuracy. Both trial and day were continuous factors. We verified that there was a significant main effect of day and reported the *p*-value.

We then calculated the Pearson's correlation of the accuracy across days and reported the correlation and significance values. Finally, we generated 1,000 distributions of correlation values with shuffled day-labels, in order to assess whether the changes we saw across days was significant given our data. We assessed this using a Monte-Carlo significance test and reported the *p*-value.

We also performed this same analysis, except using the residuals from a fitted model using all factors except day and found similar and significant results. Specifically, we took the marginal mean category decoder accuracy from an ANOVA with factors repeat number, trial, session, and mouse with only correct trials. We then performed the same analyses as stated above (Pearson's correlations and Monte-Carlo significance tests on shuffled distributions) to verify that these results were also significant.

All of these analyses were also repeated excluding lick-contaminated neurons and excluding trials where a lick comes before 600 ms post-odor onset and the results were similar and similarly significant.

## Population vector correlations

To generate population vector correlations, a form of representational similarity analysis, correlations between population vectors for each trial was computed using the corrcoef.m function. These correlation coefficients were then z-scored. To visualize within and across group differences, trials were averaged within trial types (e.g., probe versus target or probe versus all other probe trials) while never including a trial's correlation with itself.

We z-scored the population vector correlations in order to more accurately compare changes across trial types and time. By normalizing the comparisons within a session, we reduce spurious variability that can come from different numbers of neurons being recorded in each session (the variability of population vector correlations will be larger when there are fewer neurons, and smaller with more neurons if they are not normalized). We also replicated analyses in Fig 7 without normalizing the population vectors and found similar and significant results.

## Interpolating target/nontarget similarity to probe trials

To calculate the target-nontarget population (T-NT) similarity on a given trial, we took the population vector correlations between each target trial and all nontarget trials, and vice versa. Then, for each probe trial, we calculated the Probe T-NT: an average of the 10 (or another look-back window) prior trial's T-NT similarities and assigned that as the probe trial's value.

$$TNT(x) = \begin{cases} \dfrac{\sum_1^{N_{nt}} corr(x, NT_i)}{N_{nt}} & x \text{ is Target} \\[2em] \dfrac{\sum_1^{N_t} corr(x, T_i)}{N_t} & x \text{ is Nontarget} \end{cases}$$

$$Probe\ TNT(x_i) = \frac{\sum_{i-1}^{i-(L+1)} TNT(x)}{L}$$

where $L$ is the number of trials to look back (10 in main analyses), $T$ is a target trial, $NT$ is a nontarget trial, $N_t$ is the number of target trials, $N_{nt}$ is the number of nontarget trials, and $i$ is the trial number.

We compared the T-NT similarities for probe trials with those probe trials' lick latencies using a Pearson's correlation. We also performed a nonparametric significance test of that Pearson's correlation by generating a distribution of 1,000 correlations each using a different shuffled indices of latencies and T-NT similarities. We then tested whether this correlation is less than expected given the shuffled distribution by calculating the Monte-Carlo significance test.

Next, we compared these values in probe trials that were correct versus incorrect and in long or short latency probe trials using one-sided signed-rank tests. We also performed a nonparametric significance test by taking the difference in the medians (correct-incorrect) and computing this 1,000 times shuffling the accuracy values. We tested whether this difference was less than you would expect given the shuffled distribution by calculating the Monte-Carlo significance test. For the correct versus incorrect comparison, we also repeated these analyses

with look-back windows of 3 to 50 trials to look back and average over and found consistent significant results with a trial history of 9 to 16 trials.

## ANOVAs

All ANOVAs were performed using anovan.m (if more than 1 factor) or anova1.m functions. All trial or session factors were considered "continuous." To look at the residual changes across 1 factor of interest (e.g., across sessions) after accounting for the other factors in a model, we looked at the $p$-value for the factor of interest in a full model. Then we used the residuals from a fitted model using all factors except the one of interest, in order to visualize and perform post hoc statistics to describe the trend (e.g., Pearson's correlation).

## Supporting information

**S1 Fig. Different target odor mixtures.** Average choice index across the first 40 sessions for groups of mice trained on different target mixtures. Session 1 for each mouse is the first session with blocked trials. **(a)** Mouse group A trained on target mixture: Heptanal, Propyl Acetate, Isoamyl acetate. **(b)** Mouse group B trained on target mixture: Allyl buterate, Ethyl valerate, Methyl tiglate. **(c)** Mouse group C trained on target mixture: Ethyl tiglate, Allyl tiglate, Methyl tiglate. Shading represents mean +/− SEM. The underlying data for this figure are available for download from https://datadryad.org/stash/share/bC3NdXWDllJZYrRtq60q0WDQYhjZZuLulv91dm9WcYU.
(TIF)

**S2 Fig. Postmortem confirmation of tetrode placement.** This sagittal section comes from the right hemisphere of mouse S. The oval shape of the hippocampus, the shape and position of the lateral ventricle, as well as the absence of lateral olfactory tract confirm that the tetrodes were placed in the PPC and not the APC. A: anterior. P: Posterior. D: dorsal. V: ventral. Scale bar: 1 mm. DAPI staining. We did not trace the border between striatum and amygdala as we could not confidently determine it.
(TIF)

**S3 Fig. Example of spike sporting. (a)** Example of tetrode recoding. The 4 electrodes are from the same tetrode. Two single units are shown in red and blue. Grey: odor delivery. Example from mouse E, session 2-22-2017, tetrode #4. **(b)** Single unit clustering. The red and blue units are the same as (a). **(c)** Clustered single units. The red and blue units are the same as (a). The lighter traces show the 100 first extracellular action potential of each unit superimposed. The darker traces show the average extracellular action potential. **(d)** Single unit confirmation. The red and blue traces belong to the red and blue units from (a). The interspike interval analysis reveals the presence of a refractory period. The underlying data for this figure are available for download from https://datadryad.org/stash/share/bC3NdXWDllJZYrRtq60q0WDQYhjZZuLulv91dm9WcYU.
(TIF)

**S4 Fig. Lick-responsive neurons. (a-c)** Three example neurons that are modulated at first lick or reward. Raster plots (top) and PSTHs (bottom) of odor-aligned (left) or lick-aligned (right) trials. Red lines depict odor onset and offset; blue dots depict lick times. Green dots represent the end of the response period. **(d)** Proportion of the total number of neurons recorded with significant task-relevant responses. **(e)** Proportion of neurons in each session (circles) with lick-on up and lick-on down responses. Average proportions are above each group. Error bars and shading represent mean +/− SEM. The underlying data for this figure are available for download from https://datadryad.org/stash/share/bC3NdXWDllJZYrRtq60q0WDQYhjZZuLulv91dm9WcYU.
(TIF)

**S5 Fig. Odor offset responsive neurons. (a)** Two example neurons (left and right) that have both a significant odor-on up and odor-off response. Raster plots (top) and PSTHs (bottom) of odor aligned trials. Red lines depict odor onset and offset; blue dots depict lick times. **(b)** The normalized firing rate of all odor-on up and odor-off responsive neurons. **(c, d)** Same as in **a** and **b** but with odor-off only neurons. **(e)** Proportion of the total number of neurons recorded with significant task-relevant responses. **(f)** Proportion of neurons in each session (circles) with different odor-off responses. Average proportions are above each group. Error bars and shading represent mean +/− SEM. The underlying data for this figure are available for download from https://datadryad.org/stash/share/bC3NdXWDllJZYrRtq60q0WDQYhjZZuLulv91dm9WcYU. (TIF)

**S6 Fig. Controls for the proportion of modulated and selective neurons. (a)** The number of neurons recorded simultaneously in each session compared to the proportion of modulated neurons (odor-on up on the left axis, odor-on down on the right axis, Pearson's correlation). **(b)** The number of neurons evaluated for each session compared to the proportion of target-selective neurons (Pearson's correlation). **(c)** The number of neurons evaluated for each session compared to the proportion of probe-selective neurons (Pearson's correlation). **(d)** The number of neurons evaluated for each session compared to the proportion of nontarget repeat-selective neurons (Pearson's correlation). **(e, f)** We repeated comparisons while down-sampling the number of trials to control for the uneven number of trials across trial types. Details of this procedure are in the Trial down-sampled comparisons section of the methods. **(e)** Proportion of neurons significantly selective for NT repeat (black) and target (red) trials. One-sided signed rank test to determine whether there is a higher proportion of target-selective neurons than NT repeat-selective neurons separately for each of the 2 possible repeats within a session (left vs. right). **(f)** Proportion of neurons significantly selective for probe (black) and target (red) trials. One-sided signed rank test to determine whether there is a higher proportion of target-selective neurons than probe-selective neurons separately for each of the 2 possible repeats within a session (left vs. right). See methods for more details. Error bars represent mean +/− SEM. The underlying data for this figure are available for download from https://datadryad.org/stash/share/bC3NdXWDllJZYrRtq60q0WDQYhjZZuLulv91dm9WcYU. (TIF)

**S7 Fig. Average population vector corrections across groups.** Average population vector corrections for each session (open circles) between each trial type category in **(a)** overlearning sessions with at least 2 nontarget repeats, and **(b)** probe sessions. T–Target, NT–Nontarget, NT1-2 –Nontarget repeats 1–2, P1-3 –probe 1–3. Error bars represent mean +/− SEM. The underlying data for this figure are available for download from https://datadryad.org/stash/share/bC3NdXWDllJZYrRtq60q0WDQYhjZZuLulv91dm9WcYU. (TIF)

**S8 Fig. Lick latency across trial types. (a)** Distribution of all lick latencies. In subsequent analyses, licks occurring before 500 ms (1.7% of all licks) were excluded. **(b)** Trial-type lick latency differences. Marginal mean lick latency by trial type from a 3-way ANOVA with trial type, mouse, and session. Main effect of trial type: $F_{(20288,3)} = 176.3$, $P = 7.2 \times 10^{-133}$. Main effect of mouse $F_{(20288,3)} = 853.9$, $P = 0$. Main effect of session $F_{(20288,3)} = 973.2$, $P = 9.6 \times 10^{-209}$. Post hoc Bonferroni tests between trial types shown with asterisks. **(c)** Marginal mean lick latency for each mouse by trial type from a 2-way ANOVA with trial type and session. All mice showed a main effect of trial type ($P < 1 \times 10^{-20}$ in all mice). Post hoc Bonferroni tests between trial types shown with asterisks. * $P < 0.05$, ** $P < 0.01$, ** $P < 0.001$. Error bars in a represent mean +/− S.E. Note different y-axis scales. **(d)** Lick latency distributions for all trials of target,

nontarget, nontarget repeats, and probe trials, for each of the 4 mice. (**e**) Lick latencies for all target trials and (**f**) nontarget trials, separately for correct and incorrect trials. Only trials during probe sessions were used. (**g**) Left axis: A histogram of lick latencies during probe sessions by trial type. 100 ms bins. Right axis: The proportion of correct trials as a function of lick latency. Only probe sessions were used. The last data point includes latencies of 1,200–2,200 ms. Shading represents mean +/− SEM. In panels b-g, we only considered trials in which the first lick came more than 500 ms after odor onset. The underlying data for this figure are available for download from https://datadryad.org/stash/share/bC3NdXWDllJZYrRtq60q0WDQYhjZZuLulv91dm9WcYU. (TIF)

**S9 Fig. Variability in raw lick latency across sessions.** Violin plots of mouse Q's lick latencies in each session. X-ticks denote trial types. T: target; NT: nontarget; NT-R: nontarget repeat; P: probe. The underlying data for this figure are available for download from https://datadryad.org/stash/share/bC3NdXWDllJZYrRtq60q0WDQYhjZZuLulv91dm9WcYU. (TIF)

## Acknowledgments

We thank Vikrant Kapoor for help with the experimental setup, and Kenneth Blum and Gautam Reddy for feedback on the manuscript.

## Author Contributions

**Conceptualization:** Alice Berners-Lee, Elizabeth Shtrahman, Venkatesh N. Murthy.

**Data curation:** Alice Berners-Lee, Elizabeth Shtrahman, Julien Grimaud.

**Formal analysis:** Alice Berners-Lee.

**Funding acquisition:** Venkatesh N. Murthy.

**Investigation:** Elizabeth Shtrahman.

**Project administration:** Venkatesh N. Murthy.

**Supervision:** Venkatesh N. Murthy.

**Visualization:** Alice Berners-Lee.

**Writing – original draft:** Alice Berners-Lee.

**Writing – review & editing:** Elizabeth Shtrahman, Venkatesh N. Murthy.

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
