## [Editor Report · Decision Letter 0]

22 Sep 2022

Dear Dr Murthy, 

Thank you for submitting your manuscript entitled "Learning-dependent evolution of odor mixture representations in piriform cortex" for consideration as a Research Article by PLOS Biology.

Your manuscript has now been evaluated by the PLOS Biology editorial staff, as well as by an academic editor with relevant expertise, and I am writing to let you know that we would like to send your submission out for external peer review.

Once your full submission is complete, your paper will undergo a series of checks in preparation for peer review. After your manuscript has passed the checks it will be sent out for review. To provide the metadata for your submission, please Login to Editorial Manager (https://www.editorialmanager.com/pbiology) within two working days, i.e. by Sep 26 2022 11:59PM.

Kind regards,

Lucas

Lucas Smith, Ph.D.

Associate Editor

PLOS Biology

lsmith@plos.org

---

## [Decision Letter · Decision Letter 1]

14 Nov 2022

Dear Dr Murthy,

Thank you for your patience while your manuscript "Learning-dependent evolution of odor mixture representations in piriform cortex" was peer-reviewed at PLOS Biology. It has now been evaluated by the PLOS Biology editors, an Academic Editor with relevant expertise, and by several independent reviewers. 

In light of the reviews, which you will find at the end of this email, we would like to invite you to revise the work to thoroughly address the reviewers' reports.

As you will see below, the reviewers find the study to be interesting and generally well done, however they note that aspects of the study, including the task design, are not clearly presented and reviewer 3 has commented that the framing of the study needs to be revised to better reflect the state of the field. Moreover, both reviewers 2 and 3 feel additional analyses and data would be needed to better support the conclusions. We feel that these concerns preclude publication of the study in its current form, but we would welcome a revised manuscript that thoroughly addresses the reviewer comments.

Given the extent of revision needed, we cannot make a decision about publication until we have seen the revised manuscript and your response to the reviewers' comments. Your revised manuscript is likely to be sent for further evaluation by all or a subset of the reviewers.

**IMPORTANT - SUBMITTING YOUR REVISION**

*Re-submission Checklist*

*Published Peer Review*

*PLOS Data Policy*

*Blot and Gel Data Policy*

Sincerely,

Lucas

Lucas Smith, Ph.D.

Associate Editor

PLOS Biology

lsmith@plos.org

REVIEWS:

Reviewer #1: In this manuscript, the authors describe odor responses of neurons in the posterior piriform cortex and how these responses evolve when mice learn to recognize a specific combination of odors from other combinations. They find that more neurons respond to the "target" combination than other odor combinations. They also find that, with over-training, the response statistics continue to change despite no apparent change in the behavioral output. The authors find that the discriminability of target response patterns relates to the mice's ability to distinguish the specific odor combination from related but different odor combinations, suggesting a potential benefit of covert learning with over-training.

The posterior piriform cortex is still relatively poorly explored and understood. The current study, therefore, adds a significant contribution. The behavioral paradigm is clever; overall, the experiments are designed well, and the data are of high quality. The manuscript may, however, benefit from greater clarity in presentation and description.

(1) It is currently somewhat ambiguous if this study is about recognizing a specific odor combination/association with a reward (on one side) or if this is a consequence of repeated exposures. For example, previous studies on the anterior piriform cortex suggest that repeated encounters or exposure can affect the olfactory responses in this region (Schoonover et al., DOI: https://doi.org/10.1038/s41586-021-03628-7 ). In this study, the target odor combination is the most frequently encountered mixture, more than the non-target repeat odor combinations. To truly distinguish "relevance" vs. "exposure", may require a dissociation of the odor and the reward while keeping stimuli the same (e.g., Adefuin et al., DOI: https://doi.org/10.7554/eLife.76882 ). Although such an experiment is beyond the scope of this study, some clarification in the discussion may be beneficial. 

(2) In the plot presented in Figure 2b, the statistics reported show a wide distribution. Does this relate to the sample size? It may help if the number of units encountered per session is given somewhere.

(3) In calculating the correlation between population vectors, the authors seem to have normalized each unit's firing rates, normalized the correlation coefficients, and expressed them as z-scores. This is somewhat confusing and makes it harder to compare with other studies (e.g., comparison of correlation levels observed in the anterior piriform cortex e.g., Bolding and Franks, DOI: https://doi.org/10.7554/eLife.22630). Could the authors either use a simpler correlation coefficient or, alternatively, explain why their method was chosen.

(4) How does the proportion of responsive neurons relate to the correlation coefficient? Clarifying this may help relate the analysis presented in Figure 7 to the covert learning described in Fig. 5 d/e.

(5) The statement in the introduction, "... but in that case they have access to all relevant odors and do not have to depend on memory." (page 3) is unclear. Since this seems vital to explaining this study's goal, a more straightforward statement may help.

(6) The method describes the electrode placement at 3.8 mm lateral to the Bregma. All recordings presumably occurred ipsilateral to the reward port associated with the target odor combination. It may help to spell this out, even if the relationship to the tongue movement is irrelevant.

(7) Typo: 200 mm should probably read 200 um (page 26).

(8) Typo: "performed a" appears twice on page 35.

Reviewer #2: In this manuscript, Berners-Lee, Shtrahman and colleagues address two important yet poorly understood questions in olfaction: how odor mixtures are represented in the olfactory cortex, and how the representation of odors in piriform is modulated by learning and experience. 

The authors trained head-fixed mice to obtain a water reward at one lick port in response to a specific target odor mixture, and at the other lick port in response to a large panel of non-target odor mixtures. Extending this basic odor discrimination task after initial learning, experiments were also performed on overtrained mice and including non-target repeats and probe mixture trials. Tetrode recordings in the posterior piriform cortex revealed target odor mixture-specific tuning properties. Furthermore, overtraining resulted in improved odor coding and better task performance on difficult to categorize odor mixtures. 

Taken together, the authors here develop a novel odor mixture discrimination task in mice, provide a detailed description of odor mixture representations in the piriform cortex, and identify experience-dependent changes in odor representations that are consistent with improved behavioral performance. Before publication, the authors should clarify the task design and provide additional analyses to support their main findings. 

Major points: 

1) The description of the task design is confusing. For example, I suspect that basic non-target odor mixtures contain 3 components - this is not specified, and Figure 1a appears to suggest that non-target odor mixtures contain many components. How do non-target repeats differ from targets? If repeated, how do the authors distinguish between mice discriminating targets from non-targets versus mice learning about non-targets? Figure 1b appears to suggest that target and non-target trials are blocked, the methods section suggests that they are not. 

The task design is central to the entire study and its description should be clear and unambiguous for a general, non-expert audience. 

2) The data supporting improved category decoding with overlearning is difficult to evaluate. There is a small effect, supported by statistical analysis, with later sessions showing slightly improved decoding accuracy. However, day-by-day analysis reveals e.g. high accuracy in sessions 12 and 17, and low accuracy in sessions 23 and 24. Sessions 9 through 11 show consistently low accuracy and could still represent initial learning - would the statistical analysis still show a difference if sessions 12 through 28 were analyzed? Incorporating additional, more detailed behavioral measures (such as shown in Figure 5a, b) could possibly strengthen these analyses and support the authors' claim. 

3) The description of the analyses and figure panels for Figure 7 is confusing. Statements such as 'averaging the blue and green dots to get the pink dot' are not helpful. A more extensive depiction of the population vector correlation structure (such as in Figure 7c) would likely be more informative. Same trial population vector correlations of ~0.2 appear very low - please comment and clarify. 

4) What is 'overlearning'? I understand that the mice simply keep doing the task without improving their performance, which may impact odor coding and behavior. This is a potentially interesting point for further investigation, however, a more thorough introduction and discussion, for olfaction and more generally, could provide the necessary context. 

Minor points:

5) All relevant individual data points rather than the mean +/- SD only should be shown consistently, e.g. missing from Figures 1e-g. 

6) Language can be simplified throughout the description of the experiments, analyses, figure legends, and methods. The manuscript also contains a few typos and grammar mistakes that should be corrected. 

Reviewer #3: Summary

Berners-Lee et al. examined population activity in posterior piriform cortex (PPC) in head-fixed mice performing a novel 2 alternative forced choice odor-mixture task. They first show that mice can learn this complex odor-mixture task, in which a target odor mixture needs to be differentiated from a series of different non-target mixtures. They then go on to demonstrate that the PPC neurons distinguish between target and non-target mixtures more than other comparative odor-mixture pairs (non-target, probe etc). The authors interpret these results as olfactory learning-dependent signals in PPC. They also show that during over-training, where behavioral performance has plateaued, PPC ensembles continue to increase their decoding performance between target and non-target odor mixtures. Overall, the experimental results are novel, substantial, and significant. In particular, I found the quantification on the diversity of individual neuronal responses in PPC thorough and interesting. The task design is also innovative and interesting. These results will have significant implications for piriform cortex function and the field of olfactory research in general. However, there are several major concerns I have with the data, analysis, and framing. In general, I found the framing and discussion a bit lacking in reference to the state of the field and recent advances. Additional experiments, analyses, and modifications in the text should be considered and incorporated in the revision. 

Major Comments: 

1. In the results presented here, we cannot parse whether target odor mixes are better decoded from PPC because 1) it is the stimulus most frequently presented to /experienced by mice; or 2) because it is the learned target stimulus. In other words, if untrained mice were passively exposed to the same sets of odor mixtures (target, non-target, probe etc.) with the same relative frequencies without having to perform the task, would the authors observe similar differences in decoding performance in PPC? Do the authors have access to this data?

2. Related to previous point, a recent paper (Schoonover et al. 2021 DOI: 10.1038/s41586-021-03628-7) showed that repeated passive exposures to odors stabilize and reinforce odor representations in the piriform cortex. As the authors stated in the first paragraph of the first section in their discussion: "The largest over-representation was of the target odor mixture, likely showing the effect of both the repeated exposure of the odor-mixture and the reinforce dimension of the task". If the authors cannot disentangle whether the effects they see are due to learning of the task, the title ("Learning-dependent evolution…") and emphasis of the entire paper should be rephrased/reworked. This is particularly critical given the finding reported here, where odor information present in PPC population is uncorrelated with task performance. 

3. As stated in "Analysis of Behavior", trials in which mice responded before 500ms after odor onset were excluded from analysis. It is not justified why the authors chose this criterion. Please include trials in which responses were earlier than 500ms after odor onset. Specifically, 500ms is very long given what is known about the speed of olfactory decision making in rodents. See for example, Uchida and Mainen 2003 DOI: 10.1038/nn1142; Resulaj and Rinberg 2015 DOI: 10.1523/JNEUROSCI.4693-14.2015; Rinberg et al., 2006 DOI: 10.1016/j.neuron.2006.07.013). It is also not clear why mice lick with such long temporal delay as compared to previous reports. This is particularly misleading and problematic for analysis in Fig 5b. 

4. Fig 4a. It is unclear if the orange and yellow refer to modulated or selective neurons. In the figure legend, both words are used. Modulated would refer to a difference from 'baseline' (however that is defined) firing rate. Selective would refer to a test across two odor conditions (eg. between target and non-target). In this case, it is not clear which it is. If it is referring to target selective, and non-target selective neurons, what are the responses compared against? 

5. Are all trial types shown in Fig 4c presented in all sessions? Do the same data (units) go into 'target', 'probe', and 'NT repeat' analyses here? These categories do not appear necessarily mutually exclusive to each other. Is this correct? 

6. Fig 6b. What is the authors' interpretation of the fact that decoding performance is better for correct trial as compared to error trials, however there is no correlation between overall decoding and performance? I understood data from Fig. 6b to include 'overtrained' sessions. Is this correct? I would have expected no difference in decoding accuracy for correct vs. error trials for over-trained sessions. 

7. What does the PPC population activity look like during initial learning? Do the authors have access to this data? If there is no data on this, please include in the discussion a deeper consideration on what one would expect given results reported here. 

8. The authors do not sufficiently distinguish/discuss functional and anatomical differences between anterior and posterior piriform cortex (APC vs. PPC). Throughout the manuscript, results in APC were cited interchangeably as evidence in support of the authors data and interpretation. While I recognize that there have not been very many studies published on PPC besides work from the authors' group. And that results in APC could serve to motivate experiments and guide framing for results in PPC, an explicit acknowledgement of the distinction between the two should be highlighted, and discussion on the potential functional distinction between the two regions should be expanded. Currently, this is only discussed in 2 sentences in the first paragraph of Introduction on page 3. I think the authors can do a better and more thorough job on this. There are several recent findings that suggest a functional and/or anatomical difference between APC/PPC (for example, Wang et al., 2020 DOI: 10.3389/fncir.2020.00004; Chen et al. 2022 DOI: 10.1016/j.cell.2022.09.038). The authors should cite and discuss such previous work, and discuss/frame their results in context of the field. 

9. Related to previous point, there are additional gaps in citation and discussion on work in PPC which is directly relevant to the authors findings. This seems particularly unnecessary given the very few publications focused on PPC. For example: odor template matching: Zelano et al. 2011 DOI: 10.1016/j.neuron.2011.08.010, and associative learning: Poo et al. 2022 DOI: 10.1038/s41586-021-04242-3. In addition, while their results are in APC, Schoonover et al., 2021 DOI: 10.1038/s41586-021-03628-7 should also be cited and discussed in the context of the major observation of the paper (over-representation of target odor mixture). This is also relevant to the author's results in PPC decoding during overtraining. 

Minor Comments:

2. Page 4. 6th line from the bottom, the phrase "unlearned mixture" is confusing. "Unlearned" could be mis-interpreted to mean something that is first learned, and then unlearned afterwards. Consider rephrasing.

---

## [Editor Report · Decision Letter 2]

6 Mar 2023

Dear Dr Murthy,

Thank you for your patience while we considered your revised manuscript "Experience-dependent evolution of odor mixture representations in piriform cortex" for publication as a Research Article at PLOS Biology. This revised version of your manuscript has been evaluated by the PLOS Biology editors and by the Academic Editor, who has commented that the revision has largely addressed the reviewer concerns. 

Based on our Academic Editor's assessment of your revision, we are likely to accept this manuscript for publication, provided you satisfactorily address the following editorial requests.

**EDITORIAL REQUESTS: 

1) COMMENT FROM ACADEMIC EDITOR: As mentioned, the Academic Editor is largely satisfied by your response to reviewers and the changes made in this revision. However s/he has raised one lingering request. Regarding Reviewer 3's point 3 - the issue of the long latency between odor presentation and behavioral response - We think that you should add a few sentences on this topic in your manuscript, reflecting the points made in your response to reviewers.

2) ETHICS STATEMENT: Thank you for providing an ethics statement in your methods section indicating that all experiments were performed in accordance with the guidelines set by the National Institutes of Health and approved by the Institutional Animal Care and Use Committee at Harvard University. Can you please update this statement to include which specific national guidelines your protocol adhered to? We also request that you add the approval number for your animal care protocol approved by Harvard's IACUC. 

3) BLURB: Please provide a blurb which (if accepted) will be included in our weekly and monthly Electronic Table of Contents, sent out to readers of PLOS Biology, and may be used to promote your article in social media. The blurb should be about 30-40 words long and is subject to editorial changes. It should, without exaggeration, entice people to read your manuscript. It should not be redundant with the title and should not contain acronyms or abbreviations.

4) DATA AVAILABILITY: Thank you for uploading the raw underlying data for your manuscript to dryad. Can you please add a sentence to each figure legend (including supplemental) referencing this data? For example, to each figure legend you can add the sentence "the underlying data for this figure are available for download from https://datadryad.org/stash/share/bC3NdXWDllJZYrRtq60q0WDQYhjZZuLulv91dm9WcYU

5) DATA NOT SHOWN: Please note that per journal policy, we do not allow the mention of "data not shown", "personal communication", "manuscript in preparation" or other references to data that is not publicly available or contained within this manuscript. Please either remove mention of these data or provide figures presenting the results and the data underlying the figure(s). (I noticed one instance of data cited as 'not shown' on pg 38 (line 723))

We expect to receive your revised manuscript within two weeks. 

*Published Peer Review History*

*Press*

Sincerely,

Luke

Lucas Smith, Ph.D.

Associate Editor,

lsmith@plos.org,

PLOS Biology

---

## [Editor Report · Decision Letter 3]

17 Mar 2023

Dear Dr Murthy,

Thank you for the submission of your revised Research Article "Experience-dependent evolution of odor mixture representations in piriform cortex" for publication in PLOS Biology. On behalf of my colleagues and the Academic Editor, Veronica Egger, I am pleased to say that we can in principle accept your manuscript for publication, provided you address any remaining formatting and reporting issues. These will be detailed in an email you should receive within 2-3 business days from our colleagues in the journal operations team; no action is required from you until then. Please note that we will not be able to formally accept your manuscript and schedule it for publication until you have completed any requested changes.

PRESS

Sincerely, 

Lucas Smith, Ph.D.

Associate Editor

PLOS Biology

lsmith@plos.org